# Compressive instabilities enable cell-induced extreme densification patterns in the fibrous extracellular matrix: Discrete model predictions

Chrysovalantou Kalaitzidou[1]*, Georgios Grekas[2,3], Andreas Zilian[1], Charalambos Makridakis[3,4,5], Phoebus Rosakis[3,4]*

**1** Department of Engineering, Faculty of Science, Technology and Medicine, University of Luxembourg, Esch sur Alzette, Luxembourg, **2** Aerospace Engineering and Mechanics, University of Minnesota, Minneapolis, Minnesota, United States of America, **3** Institute of Applied and Computational Mathematics, Foundation for Research and Technology-Hellas, Heraklion, Greece, **4** Department of Mathematics and Applied Mathematics, University of Crete, Heraklion, Greece, **5** Department of Mathematics, MPS, University of Sussex, Brighton, United Kingdom

\* kalaitzidou.chr@gmail.com (CK); rosakis@uoc.gr (PR)

**Data Availability Statement:** The numerical code used for all simulations reported in this study is

## Abstract

We present a new model and extensive computations that explain the dramatic remodelling undergone by a fibrous collagen extracellular matrix (ECM), when subjected to contractile mechanical forces from embedded cells or cell clusters. This remodelling creates complex patterns, comprising multiple narrow localised bands of severe densification and fiber alignment, extending far into the ECM, often joining distant cells or cell clusters (such as tumours). Most previous models cannot capture this behaviour, as they assume stable mechanical fiber response with stress an increasing function of fiber stretch, and a restriction to small displacements. Our fully nonlinear network model distinguishes between two types of single-fiber nonlinearity: fibers that undergo stable (supercritical) buckling (as in previous work) versus fibers that suffer unstable (subcritical) buckling collapse. The model allows unrestricted, arbitrarily large displacements (geometric nonlinearity). Our assumptions on single-fiber instability are supported by recent simulations and experiments on buckling of individual beams with a hierarchical microstructure, such as collagen fibers. We use simple scenarios to illustrate, for the first time, two distinct compressive-instability mechanisms at work in our model: unstable buckling collapse of single fibers, and snap-through of multiple-fiber groups. The latter is possible even when single fibers are stable. Through simulations of large fiber networks, we show how these instabilities lead to spatially extended patterns of densification, fiber alignment and ECM remodelling induced by cell contraction. Our model is simple, but describes a very complex, multi-stable energy landscape, using sophisticated numerical optimisation methods that overcome the difficulties caused by instabilities in large systems. Our work opens up new ways of understanding the unique biomechanics of fibrous-network ECM, by fully accounting for nonlinearity and associated loss of stability in fiber networks. Our results provide new insights on tumour invasion and metastasis.

available in the public github repository, https://github.com/ggrekas/discrete-fiber-network.

**Funding:** CK was financially supported by Fonds National de la Recherche (FNR), Luxembourg through AFR PhD programme (AFR2020/14582656) and under the PRIDE programme (PRIDE17/12252781, DTU DRIVEN). GG was partially supported by Grant N00014-19-1-262 Vannevar Bush postdoctoral fellowship, Office of the Under Secretary of Defense for Research and Engineering. The funders had no role in study design, data collection and analysis, decision to publish, or preparation of the manuscript.

**Competing interests:** The authors have declared that no competing interests exist.

## Author summary

Living tissue often comprises a complex network of fibrous collagen and other proteins called the Extracellular Matrix, or ECM, containing embedded cells at various distances from one another. Cells can interact with their environment and communicate with peers by exerting mechanical forces onto the ECM, which acts as a medium for signalling, migration and invasion. Observed mechanical response of the ECM to forces from contracting cells is unexpected. Distinctive patterns of narrow bands emerge, where fibers are collapsed and aligned, extending unusually far into the ECM, often joining distant cells or cell clusters, such as tumours. These bands serve as conduits for mechanical signalling or highways for cell migration, tumour invasion and cancer metastasis. We hypothesise that the formation of these striking patterns results from the inability of fibers to resist compression, much like rubber bands, and the fact that in a network, there is room for fibers to rotate, buckle and collapse. Our computations successfully predict the emergent striking patterns of ECM pathways consisting of densified aligned fibers and confirm our hypothesis that multiple compressive instability mechanisms are behind the phenomenon. This brings new understanding of the role of unstable mechanical behaviour of the ECM in aiding and abetting tumour invasion and metastasis.

## Introduction

Cellular processes constitute a fundamental system of complex cascades of intracellular signalling pathways and biomechanical interactions between cells and their environment. Cells continuously remodel the extracellular matrix (ECM) through chemical and mechanical signals [1, 2]. The interplay between cells and the ECM is of great importance as it regulates a number of cellular processes [3, 4], such as cell motility and migration in physiological [5] and pathological [6–9] conditions, stem cell differentiation [10, 11], as well as cell and tissue morphology [12–14]. The intrinsic actomyosin machinery enables cells to contract, thereby exerting tractions to the fibrous ECM, which results in the generation of spatial patterns of localized deformation [13, 15–19]. Although the mechanism for their formation has not been clarified yet, there is sufficient evidence for their implication in intercellular mechanical communication. These patterns are characterized by fiber alignment and severe material densification, localized within *tethers* that join neighboring cell assemblies [12, 17], such as tumors. The tethers are densified bands where matrix density can be three to five times higher compared to the rest of the matrix. Cells were spotted to leave their cluster and advance along the axis of the tether towards the neighboring cluster [12, 17]. In addition, individual cells in collagen induced fiber alignment and network densification along lines connecting them [15], while experiments with isolated fibroblasts reported that cells grew protrusions along the generated tether and towards each other [18]. Moreover, cell-induced ECM remodeling underlies additional facets in tissue biology. Cancer studies have demonstrated the preference of tumor cells to invade along densified regions of ECM [8, 20, 21], while aligned collagen matrix serves as a highway-path which they use in order to migrate [7, 8].

In essence, every tissue component—cells and fibers—is a biomaterial with unique mechanical properties that responds accordingly to physical cues. The mechanical behavior of the ECM is thus attributed to that of its individual fibers. Previous studies [22, 23] revealed the nonlinear elastic behavior of fibers, which explains why cell-induced deformations extend substantially far and are not confined to the cell boundary [18, 24]. This nonlinearity is manifested

by strain stiffening in tension [23, 25–28] and buckling in compression [16, 18, 29–31]. These effects have been extensively investigated before [18, 32–40], wherein fibers are modeled as homogeneous beams with stable (monotonically increasing) force-stretch responses. This would seem to suggest that the whole matrix (fiber network) would exhibit stable mechanical behavior. It might come as a surprise that this is actually not true even in the case of non-buckling fibers with a linear force-stretch response: in [41] it is shown that the energy of such a network is a nonconvex, multi-well function of nodal displacements. This occurs because fibers can undergo large rotations [41], and implies instability. In our recent work [19, 42], instability is also encountered in a continuum model, obtained from orientational averaging of individual fiber response, even when the latter is stable. This model predicts localized densification patterns that bear a strong resemblance with experiments [13, 17, 19].

Subjected to large strains, fibers can be extremely extensible without breaking [43] and they stiffen with increasing tension [26, 27]. Especially interesting is the case when fibers are subjected to compressive forces under which they buckle, losing stiffness and eventually collapsing. Experiments and analysis of open-cell foams, essentially elastic fiber networks [44], revealed multiple regimes of stress-strain response marked by a non-linear softening instability in compression, coupled with network densification. The cause of the instability was identified as buckling of fibers or polyhedral fiber elements [44]. Similar conclusions with emphasis on instabilities were also drawn for elastoplastic honeycombs [45] and later for fibrin networks [46]. Rather than homogeneous rods, ECM fibers have a bundlelike morphology characterized by a complex hierarchical structure [28, 43]. This assembly gives rise to unexpected mechanical effects. Recently, uniaxial compression experiments and simulations of hierarchical beams [47] revealed a post buckling response that is highly unstable compared to that of homogeneous beams. It involves a transition from hardening (positive slope in load-displacement response) to softening (negative slope) with increasing compression, while they were reversible upon load release. Similar behavior was observed in [48], carbon nanotubes [49] and architected rods [50]. These studies suggest that the mechanics of single collagen fibers is more complicated, exhibiting unstable behavior that deviates from what has been addressed so far. Indeed, experiments on buckling of single collagen fibrils [51, 52] reveal some post-buckling shapes that are inconsistent with the traditional stable (linear or stiffening) models of beams, but are predicted by theoretical models assuming instability [53, 54]. Here, we incorporate such a fiber instability as a model hypothesis and compare its consequences to more traditional stable-fiber models, in an effort to understand ECM-related deformations.

Accordingly, we introduce two different families of network models, with different fiber force-displacement relations modelling the post buckling behavior of fibers, exhibiting distinct nonlinearity and stability features. Family 1 displays a positive but decreasing stiffness with increasing compression, while Family 2 entails an instability phase, where stiffness becomes negative at extreme compression, invoking recent experimental and theoretical evidence on unstable buckling of hierarchical beams, including collagen fibers [47–54].

Buckling is a bifurcation [48]. At a critical force, the single equilibrium of a beam or structure branches into multiple equilibria, some of which are stable and some unstable. There is actually stable and unstable buckling, corresponding to supercritical and subcritical bifurcations, respectively; see [48] for a clear explanation. In stable buckling, the system buckles smoothly and gradually as forces are increased, while maintaining stability. A single Family-1 fiber is an example, representing the traditional view of stable (supercritical) post-buckling behavior. In unstable buckling, the system becomes unstable, and must dramatically jump over unstable states to a far-away stable equilibrium. This occurs for a fiber of Family 2, modelling buckling of hierarchical beams [47–54] that shows clear evidence of unstable (subcritical) buckling. We identify two distinct instability modes for fiber networks here. The first is

unstable buckling of single fibers (Family 2). The second occurs even when the fibers are stable (Family 1) but whole elements (larger fiber groups, triangles in our model) suddenly become unstable, buckle and collapse. The typical example is snap-through buckling in simple trusses [48]. The present work is the first to consider both stable and unstable buckling, and the possibility of fiber group unstable collapse, as distinct mechanisms in biological fibrous networks. All these instabilities occur due to compressive forces, and lead to large increases in density. Parts of this work have appeared in [55].

We perform extensive simulations of a fiber network containing one or more contracting circular cells, in the spirit of [18]. Simulations with our Family-2 models predict ECM densification, the formation of experimentally observed densified tethers between pairs of contracting cells and enhanced fiber alignment localized within the tethers. We show that ECM densification and fiber alignment are simultaneous consequences of fiber compression instability inherent in Family 2. These phenomena are stronger and highly localized within more sharply defined zones, compared to predictions of previous models with stable Family-1 type behaviour [18, 33–40]. Surprisingly, densified zones with aligned fibers also appear in Family-1 simulations, albeit with important differences. First, they require much higher levels of cell contraction and second, they reveal a different instability mechanism, namely snap-through buckling of larger fiber groups, such as triangular elements. This distinct compression instability mechanism is also evident in simulations with the linear model and has not been addressed by previous studies involving discrete networks. These observations show that compression instabilities are ubiquitous in elastic fiber networks, and their presence is critical and essential for localized densification and fiber alignment.

## Methods

We have developed a 2D discrete model of a fiber network representing the natural ECM to explore its mechanical behavior under cell induced loading. In particular, we partition a circular domain into triangular elements. Each of the three sides of an element represents an individual fiber Fig 1a. Triangle vertices are the nodes of the network, where fibers terminate, so that *fiber length* corresponds to the length of the segment between two nodes. The nodes are modeled as movable frictionless hinges with two degrees of freedom. In general, nodal displacements change the length of fibers, which are modeled as nonlinear elastic springs. As a result, the total energy of the network (sum of individual spring energies) is a function of all nodal displacements. This leads to the problem of minimizing the total energy over nodal displacements. This problem is solved numerically using advanced optimization techniques (see subsection *Formulation, Software and statistical analysis* below).

## Mechanical properties

For a single fiber, we introduce the *effective stretch* λ, which equals the distance between its endpoints divided by its undeformed, or relaxed, length. The energy of a single fiber can be written as $W(\lambda)$ as a function of effective stretch λ. When the fiber is in tension, it is straight and λ equals the actual stretch (strain +1), while $W(\lambda)$ equals the elastic energy due to stretching of the fiber. When the fiber is in compression, it may be buckled, in which case the elastic (mostly bending) energy of the fiber can still be expressed as a function $W(\lambda)$ of the distance between its endpoints, hence of the effective stretch λ. See Fig 1b. In that case $W(\lambda)$ is chosen to embody the post-buckling response of the fiber. If the deformed-position vectors of the fiber end points (nodes) are $\mathbf{x}_i$ and $\mathbf{x}_j$ and the undeformed fiber length is $l_{ij}$, the energy of the

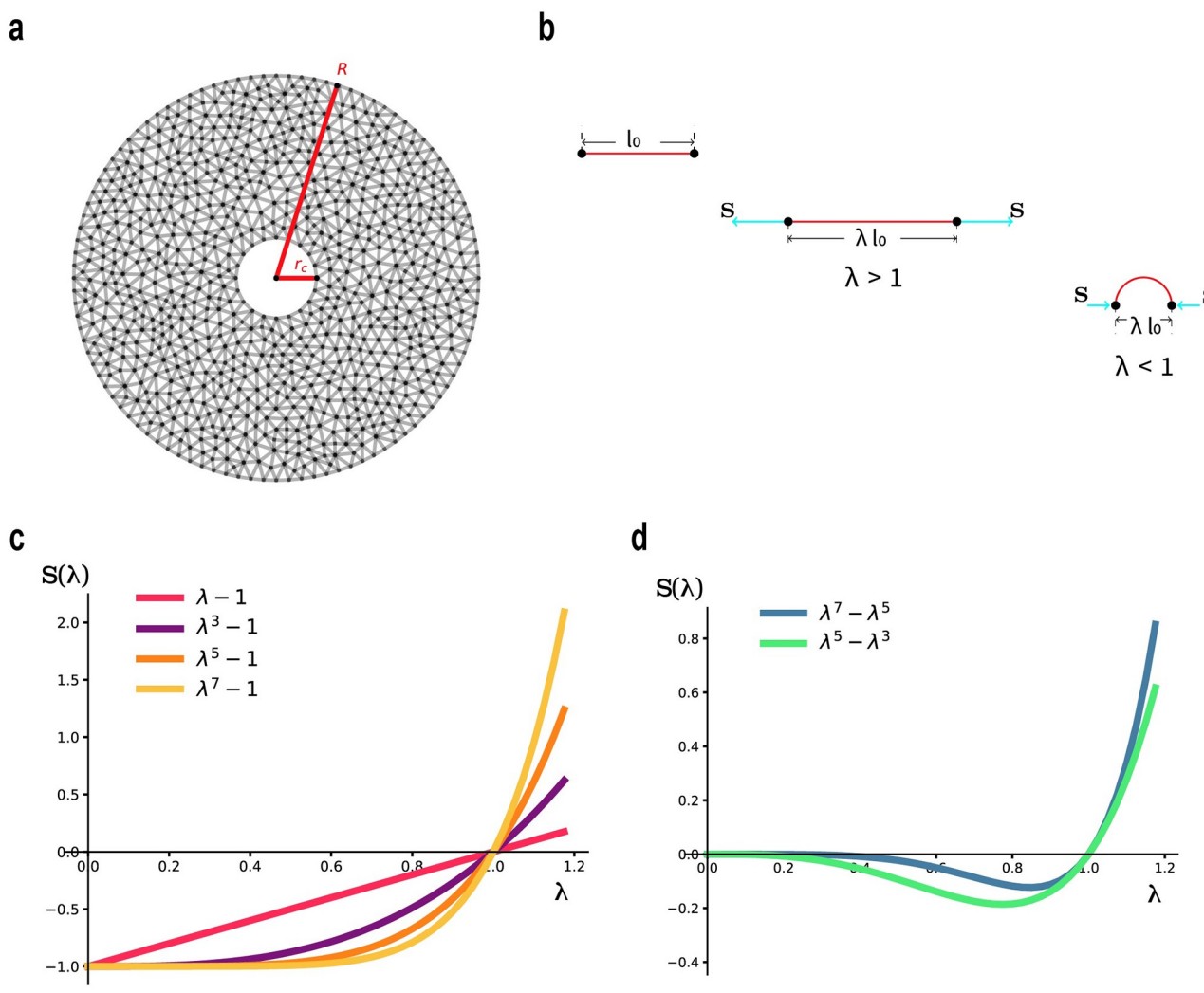

**Fig 1. Model description. (a)** Example of our 2D discrete fiber networks. Each edge corresponds to an individual fiber. The cavity represents a cell with undeformed radius $r_c$. **(b)** Effective stretch $\lambda$ of a single fiber. Here $\lambda$ is defined as the ratio of deformed to reference ($l_0$) distance of a fiber's endpoints. From left to right: a relaxed fiber with length $l_0$, a fiber under tension ($\lambda > 1$) and a buckled fiber under compression ($\lambda < 1$). The cyan arrows represent the applied loads at the fiber's endpoints. **(c,d)** Force-stretch ($\lambda$) curves of individual fibers. **(c)** Family 1: $S_{1k}(\lambda) = \lambda^k - 1$, $k = 1, 3, 5, 7$ **(d)** Family 2: $S_{2k}(\lambda) = \lambda^k - \lambda^{k-2}$, $k = 5, 7$.

fiber is

$$W\left(\frac{|\mathbf{x}_i - \mathbf{x}_j|}{l_{ij}}\right), \tag{1}$$

the quantity within parentheses above being the effective stretch $\lambda$ of the fiber between nodes $i$ and $j$. We explore various models of the mechanical behavior of fibers, characterized by the force-stretch relation of a single fiber $S = S(\lambda)$, where

$$S(\lambda) = dW(\lambda)/d\lambda, \tag{2}$$

is the fiber force, nondimensionalized after dividing by a coefficient with dimensions of force. These models were designed to capture fiber stiffening (increase in the stiffness $dS/d\lambda$) in

tension ($\lambda > 1$) [26, 27], but also softening (decrease in stiffness $dS/d\lambda$) in compression as $\lambda <$ 1 decreases toward 0. Softening is attributed to buckling [18, 19, 32]. For direct observation of buckled fibers in collagen networks see [29, 31]. We introduce two families of models. Family 1, shown in Fig 1c,

$$\text{Family 1}: \quad S = S_{1k}(\lambda) = \lambda^k - 1, \quad k = 1, 3, 5, 7 \tag{3}$$

which includes the linear case (k = 1) and Family 2 shown in Fig 1d:

$$\text{Family 2}: \quad S = S_{2k}(\lambda) = \lambda^k - \lambda^{k-2}, \quad k = 5, 7. \tag{4}$$

While for all models there is stiffening in tension (except for the linear one $S_{11}$), the difference between the two constitutive families is in compression. In all nonlinear models in Family 1, force $S$ and stiffness $dS/d\lambda$ both increase monotonically with increasing stretch $\lambda$, Fig 1c (except for the linear one $S_{11}$, where stiffness remains constant). However, stiffness decreases monotonically with compression (as $\lambda$ decreases to zero) until it vanishes in the crushing limit $\lambda \to 0$, whereas force reaches a plateau where it remains approximately constant as $\lambda \to 0$. Models differ in how abrupt the loss of stiffness is and at which level of compressive stretch it occurs. Thus Family-1 fibers can sustain a limited amount of compressive force even after buckling. See also [19] where similar fiber behavior is used to derive a continuum model. In contrast [18] and [40] use a bilinear model with piecewise constant stiffness that is lower in compression. Partial loss of stiffness in Family-1 type force response is intended to model the post buckling behavior (load versus distance between ends) of nonlinear elastic bars in compression by axial loads in the context of large deformations. Family-1 behavior is consistent with experiments and simulations of post-buckling in certain homogeneous nonlinear elastic beams [50]. Thus Family 1 corresponds to stable (supercritical) post-buckling behavior [48]. The linear model with its constant positive stiffness is an exception, intended to model fibers that do not buckle in compression.

### Evidence for Family-2 unstable fiber buckling model

Actual ECM fibers are far from homogeneous, but exhibit a bundle-like morphology with a complex hierarchical structure [28, 43, 51, 52]. This gives rise to unexpected mechanical effects. In a recent study [47] with hierarchical beams, uniaxial compression experiments and simulations revealed a post-buckling response where the force-stretch relation changes from positive to negative stiffness (slope) for high enough compression. This also occurs in beams composed of metamaterials [50] and in carbon nanotubes [49]. Additional evidence for the unstable buckling of collagen fibers comes from an anonymous reviewer who brought experimental studies of single collagen fibril buckling [51, 52] to our attention. These studies point out that a linear moment-curvature relation predicts a sinusoid as the buckled shape for these fibrils on an elastic pre-stretched substrate. However, in many experiments reported in [51, 52], the observed shape is not sinusoidal, but exhibits strong curvature localizations (see Figs 3e, 3g and 3h in [51]) and corners (sawtooth shape, see Fig 3d in [51]). These are inconsistent with the linearity assumption, as noted in [51]. They are nonetheless consistent with analytical predictions of nonlinear models [53, 54] that assume an unstable regime with decreasing moment-curvature relation. In particular, the analysis in [54] leads to curvature localizations (assuming eventual restabilization at large curvature), while [53] predicts a sawtooth shape with corners in case the moment keeps decreasing with curvature. Moreover, the description of fibril microstructure in [52] is topologically similar to that of hierarchical beams studied in [47], which also develop corners in buckled shapes (see Fig 10b in [47]) and exhibit an unstable negative slope in stress-effective stretch for high enough compression (see Fig 10a in [47]).

Family-2 models were designed to capture this subcritical (unstable) buckling behaviour (Fig 1d). Initially positive stiffness (slope) decreases monotonically with decreasing $\lambda < 1$ as the fiber initially resists the compressive load, after which stiffness becomes negative with further compression, entering a compression instability regime (negative stiffness) up to final collapse as $\lambda \to 0$.

### Energy with penalty for interpenetration of matter

Suppose one of the models from (3) or (4) has been chosen. The corresponding elastic energy of a fiber is then given by

$$W(\lambda) = \int_1^\lambda S(\gamma)\, d\gamma \qquad (5)$$

with a minimum at $\lambda = 1$ when the fiber is unstretched. Let $\lambda_j$ be the stretch of fiber j, where $j = 1,2,\ldots, F$ and $F$ is the total number of fibers in the network. Therefore, the total fiber network strain energy is equal to

$$E(\mathbf{x}_1, \ldots, \mathbf{x}_N) = \sum_{k=1}^{F} W(\lambda_k) = \sum_{i=1}^{N}\sum_{j=1}^{N} k_{ij} W\left(\frac{|\mathbf{x}_i - \mathbf{x}_j|}{l_{ij}}\right). \qquad (6)$$

Here $F$ is the number of fibers, $N$ the number of nodes and $k_{ij} = 1$ if there is a fiber joining nodes $i$ and $j$, 0 otherwise.

Our initial attempt is to impose displacements or applied forces on boundary nodes of the network and minimize the total network energy $E(\mathbf{x}_1, \ldots, \mathbf{x}_N)$ with respect to all positions of interior nodes $\mathbf{x}_i$. However, allowing for large contractile deformations can result in nonphysical solutions due to interpenetration of matter. This happens when triangular elements fold over and snap-through to the other side, as there is no resistance against fibers crossing through each other in the model. Examples are shown in Supporting information, S1(I) Fig. This is an instance of the well-known snap-through buckling subcritical instability of structural mechanics, e.g., [48, 56] and shows that the energy $E$ is nonconvex and likely to have multiple local minima, as well as unstable saddle points and local maxima, even in the case of the linear fiber model $S_{11}(\lambda) = \lambda - 1$ as schematically shown in Fig 2b and 2c. This and other instabilities also occur in a regular (lattice) network of linear springs [41].

Solutions with snap-through buckling involve interpenetration of matter and orientation reversal, which are both physically unacceptable. In order to exclude such solutions, an energy penalty term is introduced that resists any two fibers with a node in common from crushing into each other. That would be equivalent to the oriented area of the associated triangular element going to zero, then becoming negative with orientation reversal. Interpolating the deformation from nodes to the entire domain in a piecewise affine way (continuous overall and linear in each triangle), we define $J$ to be the Jacobian determinant of the deformation. Hence $J$ is piecewise constant and equal to the *ratio of deformed to undeformed oriented triangle area*. Letting $\mathbf{x}$ and $\mathbf{z}$ be two undeformed vector sides of a triangle, and $\bar{\mathbf{x}}, \bar{\mathbf{z}}$ be the deformed sides, we have

$$J = \frac{(\bar{\mathbf{x}} \times \bar{\mathbf{z}}) \cdot \mathbf{k}}{(\mathbf{x} \times \mathbf{z}) \cdot \mathbf{k}} \qquad (7)$$

in terms of the vector product, where $\mathbf{k}$ is the out-of-plane vector. Negative $J$ denotes orientation reversal of the respective elements, i.e. folding over and interpenetration of matter. The

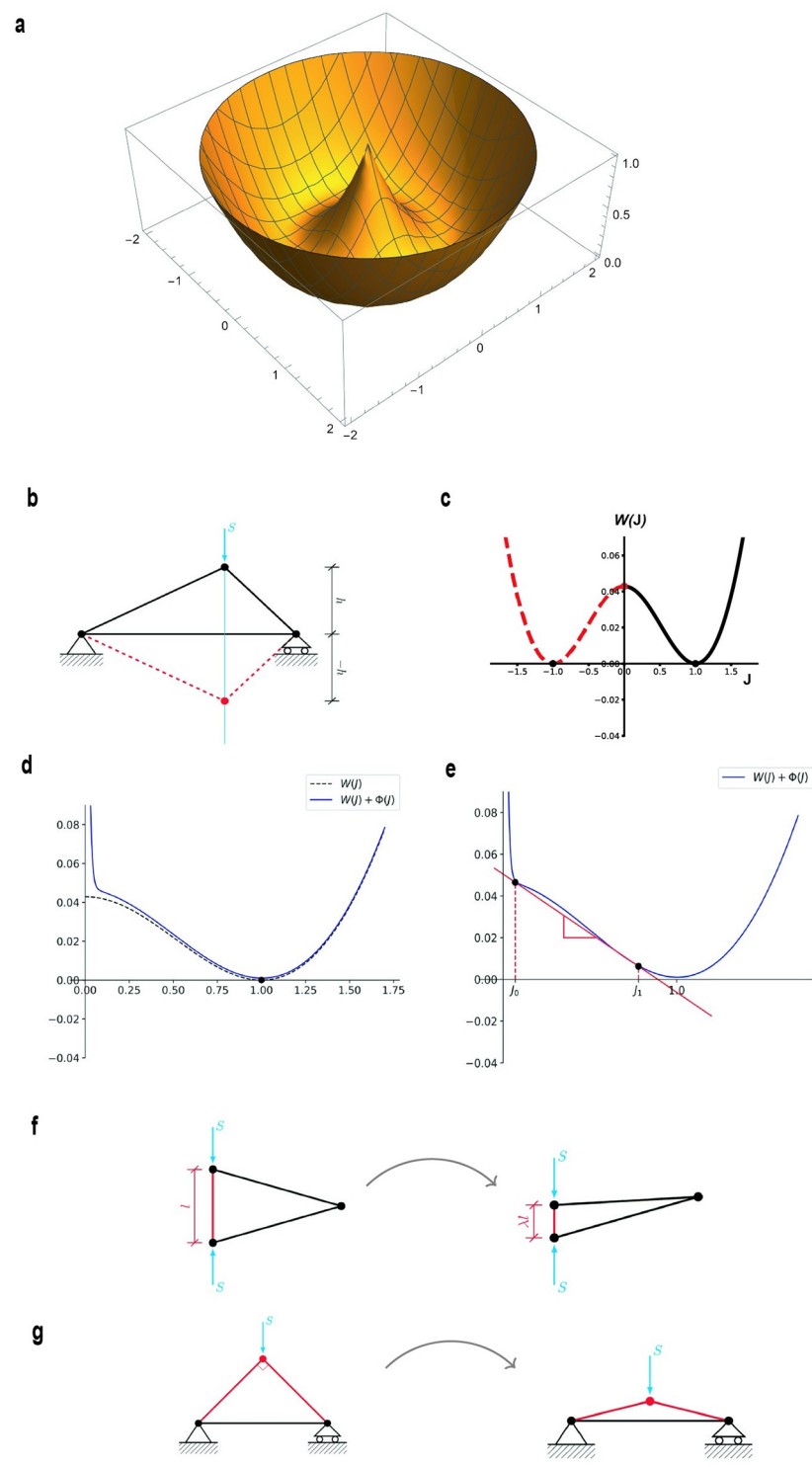

**Fig 2. Instability mechanisms. (a)** Energy (1) of single linear fiber with one end $\mathbf{x}_j$ fixed at the origin, as a function of the position vector $\mathbf{x}_i$ of the other end (nonconvex surface of revolution of a convex parabola with minimum at 1). **(b)** Element collapse Instability (snap-through of triangular elements) under compressive force $S$ (cyan) **(c)** Energy of a triangular element as a function of its oriented area ratio $J$. Note nonconvexity and two-well structure. **(d)** Dotted line: as in (b) for $J > 0$. Solid line: energy with penalty $\Phi(J)$ added. **(e)** Penalized energy has two stable equilibria $J_0$ and $J_1$ under suitable compressive force (equal to the slope of the red straight line) **(f)** Fiber collapse (red fiber) causes triangular element area collapse. **(g)** The converse is not true. Triangular element collapse can happen without fiber collapse.

penalty term is chosen as a function of $J$,

$$\Phi(J) = e^{-Q(J-b)} \tag{8}$$

where $b > 0$ is a small constant and $Q > 0$ is large (see Supporting information, S1II Fig, we choose $Q = 50$ and $b = 1/4$. We find results are qualitatively similar for various values of these parameters, see Supporting information S2 Fig). Thus $\Phi(J)$ is very small for $J > b$ and becomes very large for $J < 0$. Hence it serves in maintaining positive orientation in the network, as negative orientation ($J < 0$) is costly in energy. For elements with positive area ratio, $\Phi(J)$ is very small, thereby having essentially no contribution to the network's total energy. Physically it corresponds to fibers (which actually have nonzero thickness) resisting being crushed together when network elements are close to collapsing and neighboring fibers come into contact. The modified network potential energy has the form:

$$\hat{E} = \sum_{j=1}^{F} W(\lambda_j) + \sum_{k=1}^{K} A_k \cdot \Phi(J_k) \tag{9}$$

where $F$ is the total number of fibers in the network, $K$ the total number of elements, $W(\lambda_j)$ the potential energy of an individual fiber, $A_k$ the reference area that element $k$ occupies and $J_k$ its oriented area ratio.

## Nonconvexity and instability

Stability of equilibria is closely related to convexity of the energy as a function of nodal positions. Equilibria are stable if they correspond to local minima of the energy, where the latter is necessarily locally convex. In contrast, critical points where the energy loses convexity correspond to unstable equilibria. Even after modifying the energy by adding the penalty term, instabilities due to nonconvexity are still present. We identify some instability modes next.

*Nonconvexity due to Large Rotations*

Even if the single fiber energy $W(\lambda)$ from (5) is strictly convex with a minimum at $\lambda = 1$, the corresponding energy in (1) is a nonconvex function of nodal position vectors $\mathbf{x}_i$ because of rotational invariance (Fig 2a, see also Fig 4 in [41]). This occurs even for the linear fiber model ($k = 1$ in (3)). It is a source of nonconvexity of the total energy $\hat{E}$ related to geometric nonlinearity, and is typically entirely missed when small displacements/rotations are assumed.

*Element Collapse Instability*

Before a triangular element consisting of three fibers (Fig 2b) undergoes snap-through (its oriented area changes signs) it buckles, or collapses, when a node touches the opposite side. This is actually an unstable equilibrium of the triangle energy, which is thus a nonconvex function of element oriented area ratio $J$. Compressing an element triangle along its height (Fig 2b), we observe that the energy of the triangle as a function $W(J)$ has minima (and vanishes) at $J = 1$ and at $J = -1$ (Fig 2c) after the triangle has snapped-through to its mirror image. Since $W(J)$ is odd it must be nonconvex with an unstable equilibrium at $J = 0$. In our model, such total collapse is prevented by the penalty term $\Phi(J)$, but bistability and noncovexity of the penalized energy $W(J) + \Phi(J)$ remains (Fig 2d), with an additional, highly compressed solution for some values of compressive force (Fig 2e). This occurs in both model Families, even in the linear model; it is an example of the well-known snap-through instability of structural mechanics, e.g., [48, 56]. In order to identify this instability mode in our simulations, we define the *densification ratio* $\varrho$ of each triangular element to be the ratio of deformed to reference density of a hypothetical continuum deforming as the triangle. This gives

$$\varrho = 1/J. \tag{10}$$

We observe that the snap-through element buckling just described is attributable to the loss of convexity due to large rotations discussed before. The energy of a triangular element with fixed base as a function of the opposite nodal position is a sum of two sombrero-like energies (Fig 2a) with different centers and has precisely two global minima giving rise to the bistable form in Fig 2c. Snap-through involves an unstable subcritical bifurcation. Eventually the penalty term (8) and (9) restabilizes the element against total collapse. An unstable regime remains in general, rendering fiber compression response essentially bistable, Fig 2e.

*Fiber Collapse Instability*

In Family-2 networks there is an additional instability: when a fiber is compressed past the point where the slope of the $S(\lambda)$ curve becomes negative (Fig 1d), it enters an unstable regime, tending to collapse to zero effective stretch. For example, the unstable regime for $S(\lambda) = \lambda^7 - \lambda^5$ is $0 < \lambda \leq 0.85$. Clearly, fiber collapse would imply area collapse of any element (triangle) with this fiber as a side (Fig 2f). Eventually the penalty term (8) and (9) restabilizes the element against total collapse, since collapse of one of the sides of a triangle implies area collapse, activating the penalty term.

To summarize, all fiber networks are susceptible to element collapse (triangle snap-through buckling) instability. Family-2 networks suffer from an additional fiber collapse instability brought about by total loss of strength due to buckling of hierarchical fibers. Simple geometry shows that fiber collapse implies element collapse (Fig 2f), but not vice versa (Fig 2g).

## Formulation, software and statistical analysis

By expressing stretches $\lambda_j$ and Jacobians $J_k$ in terms of variable nodal positions $\mathbf{x}_i$, $i = 1, \ldots, N$, we express the total energy $\hat{E}$ in (9) as a function $\hat{E}(\mathbf{x}_1, \ldots, \mathbf{x}_N)$ of nodal positions. See (6) for the first term. The boundary conditions are applied on the cell-boundary nodes and simulate cell contraction, as described below in *Model Geometry*. We then perform energy minimization on $\hat{E}$. For the energy minimization procedure the Nonlinear Conjugate Gradient method has been employed.

The discrete model has been implemented in Python [57]. The triangulation has been implemented in FEniCS [58] and the optimization method (for finding energy minima) is explained in [19]. The statistical analysis has been done in R [59] and for multi-group comparisons we used one-way analysis of variance (ANOVA).

## Model geometry

Cells are modeled as circular cavities of radius $r_c$ within the domain of outside radius $R$. Thus, the domain containing the ECM network is an annulus with $r_c < r < R$, where $r = |\mathbf{x}|$ is the radial distance from the domain center and $\mathbf{x}$ is the position vector. In particular, we model contractile cells. We simulate contraction by prescribing an inward radial displacement of cell boundary nodes given by:

$$\mathbf{u}(\mathbf{x}) = -\frac{u_0}{r_c}\mathbf{x} \tag{11}$$

for nodes with position vector $\mathbf{x}$ such that $|\mathbf{x}| = r_c$. Simulations with two cells involve two distinct cavities with contractile displacement applied on the boundary of each. The outer boundary $|\mathbf{x}| = R$ of the network is free (no applied forces or prescribed displacements) in all simulations.

## Results

### Single-cell simulations

**Severe localized densification patterns are observed in Family-2 models, moderate ones for Family 1.**   We simulate a single cell contracting within a fibrous network for each one of the models introduced in (3) and (4). Simulations at 50% cell contraction exhibit patterns of highly localized, severe densification shown in Fig 3a–3c and S3a–S3c Fig. These patterns take the form of bands, emanating from the periphery of the contracting cell into the surrounding matrix. Plotting the densification ratio of each element versus distance from the cell shows that in Family-2 models, highly densified elements have densification ratio $\varrho \approx 3$ and reach up to six cell radii into the ECM (Fig 3c). In contrast, Family-1 densified triangles are confined within two cell radii (Fig 3a–3b), with densification ratio $\varrho$ at most 2.

The distribution of fiber stretches within the deformed networks illustrates similarities and differences between models (Fig 3d–3k and S3 Fig). Fibers under tension ($\lambda > 1$) align roughly with the radial direction, forming continuous paths that propagate a few cell diameters into the matrix (Fig 3d–3f and S3d–S3f Fig). This happens regardless of the model, though in Family-2 simulations the paths extend further into the matrix (Fig 3f and S3f Fig). When it comes to compressed fibers, things differ significantly between models (Fig 3g–3k and S3g–S3k Fig). Fibers under compression ($\lambda < 1$) are oriented close to the angular direction, forming loops around the cell (Fig 3g–3k). Within each of these loops, and close to the cell boundary, the stretch is nearly uniform for Family-1 models (Fig 3g–3h). Similar behavior is seen in [40, Fig 6].

Simulations with Family 2 exhibit two differences: the distribution of compressive stretch around the cell is strongly inhomogeneous (Fig 3k), and the maximum compression is up to twice as high as in Family-1 simulations, 60% compressive strain (or stretch $\lambda \approx 0.4$) compared to 30% ($\lambda \approx 0.7$) for Family 1 (colorbars in Fig 3g–3k and S3g–S3k Fig). Compressed Family-2 fibers are still roughly in the angular direction. The most compressed fibers occur within narrow bands emanating radially from the cell and reaching as far as 6 deformed cell radii into the matrix (Fig 3k–3m, S3k and S3n Fig). Furthermore, in Family 2, network triangles comprising the densified bands are excessively compressed (Fig 3m), as they contain fibers that have nearly collapsed. Fibers under tension are aligned along the axis of densification bands, roughly perpendicular to fibers under compression (Fig 3n). When the densification ratio of the networks in Fig 3c is compared to the compressed fiber distribution of Fig 3k, it becomes clear that regions of localized excessive densification ($\varrho \approx 3$) coincide with the bands containing severely compressed fibers (Fig 3c and 3k–3n, S3c and S3k Fig).

In Family-1 simulations, severe compressive stretch is not observed at 50% contraction level, with $\lambda$ remaining above 0.7, compared to 0.4 for Family 2. Densified zones are much shorter and confined to the immediate vicinity of the cell (Fig 3a–3b and S3a and S3b Fig) with triangles less compressed ($\varrho$ at most 2 compared to 3 for Family-2).

**Size effects.**   There are at least two characteristic lengths in this problem: Fiber length (characteristic network size) and particle diameter ("particles" refer to individual cells [18], multicell clusters such as mammary acini [17], or artificial contracting particles [19]). The ratio of fiber to particle size has a major effect on the extent of densification patterns. Around single fibroblasts, just a couple of densified protrusions are typically observed [18], whereas around much larger contracting particles compared to network fiber size, at the same radius contraction ratio, a much higher number of thin radial hairlike bands occurs [19], to the effect that it is sometimes difficult to discern individual radial hairs without close inspection, and they appear to almost form a homogeneous "halo" around the particle, e.g., Figs 4a, 4d, 5b and 6b in [19]. Our simulations capture this size effect: S4 Fig, panel (a) with particle size a few

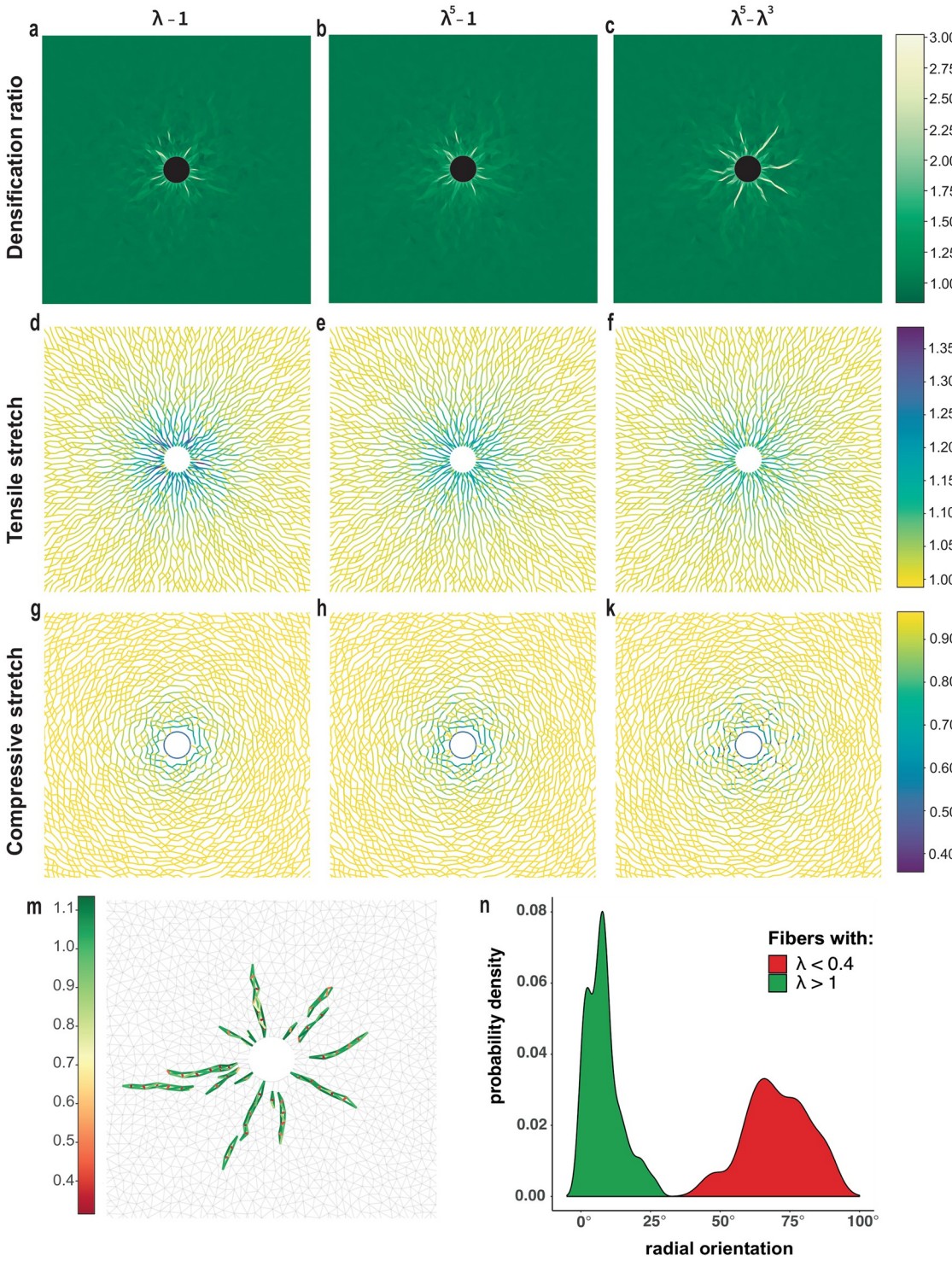

**Fig 3. Fiber collapse instability and severe localized densification.** Simulations with a single cell at 50% contraction with Family-1 models $\lambda - 1$ and $\lambda^5 - 1$ and Family-2 model $\lambda^5 - \lambda^3$ **(a-c)** Densification ratio of triangular elements (color plot) in deformed networks **(d-f)** tensile stretches and **(g-k)** compressive stretches in deformed fibers **(m)** stretch of deformed fibers and **(n)** radial orientation distribution of fibers within the densified bands in Family-2 case. Colorbars: (a-c) densification ratio $\rho$ of the deformed networks, (d-k) fiber stretch $\lambda$.

times larger than fiber size shows just two densified protrusions with densification ratio above 2.4 reminiscent of single cell behaviour. At the same contraction level, particles 20 times larger compared to fiber size produce many extended fine hairs with stronger densification ratio exceeding 3 (S4 Fig, panel (b)). Both simulations used the same Family-2 5–3 model and only mesh size differs between them. In S4 Fig, panel (c), the same network, particle and contraction level as in panel (b) is used, but with Family-2 model 5–7, producing roughly the same number and fineness of fibers (but they are substantially longer than in the 5–3 model simulation S4 Fig, panel (b)). We also note that simulated densification patterns in this work show area densification ratio (10), whereas experimental images in [19] depict fiber density ratio; see Supporting Information, S5 Fig for an explanation.

The size effect discussed above is due to discreteness of the network model. In the continuum model of [19], capturing size effects requires adding strain gradient terms to the nonlinear strain-dependent elastic energy.

**Comparing critical levels for densification pattern formation in the two model families.** Since the main difference between the two model families is the unstable compression regime in Family-2 models, where the $S(\lambda)$ curve has negative slope (Fig 1d), the results above suggest that fiber collapse instability is responsible for the sudden growth of localized densification bands. In order to test this hypothesis, we simulate network response under gradual cell contraction, to study the onset of densified band formation.

Fig 4a–4e shows a Family-2 network with a cell contracting at five consecutive levels, from 20% to 40% reduction of cell initial radius. Initially, as contraction progresses, the densification ratio of essentially the same few triangles proximal to the cell increases linearly with cell contraction (Fig 4a–4d) up to 35%. Remarkably, at the next level of (40%) contraction, densified bands around the contracting cell have appeared, extending noticeably further into the matrix (Fig 4e). Below each plot of Fig 4a–4e, in a "tree diagram", we plot the stretch of each individual fiber (abscissa) versus distance from the cell center (ordinate) for each contraction level; color indicates orientation relative to the radial direction. The evident asymmetry near the base of each tree at larger contractions shows the difference of compressive versus tensile stretches. Tensile stretches $\lambda > 1$ grow gradually with increasing contraction. In fibers under compression ($\lambda < 1$), the stretch first decreases slowly, with only a few fibers in the unstable regime $\lambda < 0.85$ (red dotted line), all of whom are close to the cell up to 30% contraction. At 35% there is a steep increase in the number of fibers below the threshold, with stretches down to 0.4 and reaching more than 6 cell radii into the network by 40% contraction. Going back to the respective densification ratio configurations, we observe that the jump in fiber compressive stretch and the abrupt appearance of densified bands occur at the same contraction level between 35% and 40%.

This trend in densification localization is reflected in plots of the maximum over the network of the densification ratio inverse $1/\varrho_{max}$, and the minimum stretch $\lambda_{min}$, at each contraction level (Fig 4f). We note that $1/\varrho_{max} = J_{min}$, the minimum area ratio, corresponding to the most compressed triangular element. The densification ratio first increases slowly with contraction, then there is a steep rise between 35 and 40% contraction, the level at which extensive localized densification is spotted (Fig 4e). The minimum fiber stretch follows exactly the same behavior as $1/\varrho_{max}$, the two curves in Fig 4f being nearly identical. Initially, $\lambda_{min}$ decreases approximately linearly with contraction, namely the minimum stretch occurs at the cell boundary as dictated by the boundary conditions

$$\lambda_{min} = 1 - \gamma, \tag{12}$$

with $\gamma$ the fractional cell diameter decrease. Then there is a sudden drop in stretch magnitude

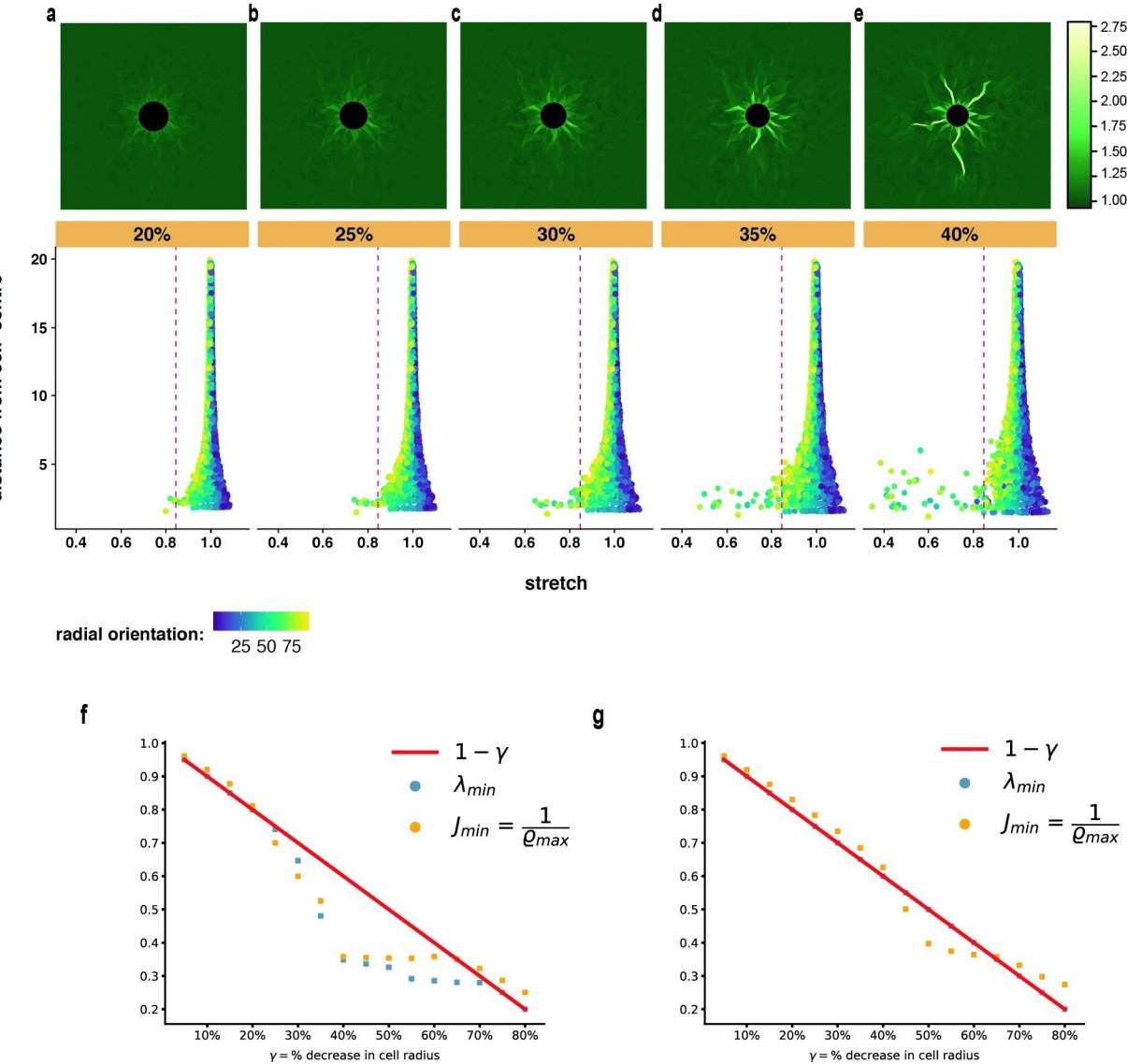

**Fig 4. Progressive cell contraction and densification localization. (a-e)** Simulations with Family-2 model $S(\lambda) = \lambda^7 - \lambda^5$ of a cell contracting in the range 5%–80%. Top: densification ratio $\varrho$ color plot at each indicated contraction step. Bottom: tree diagrams, fiber distance from cell center versus fiber stretch for all fibers in the network at each contraction step, *x axis*: fiber stretch, *y axis*: fiber distance from cell center. **(f,g)** Maximum densification ratio inverse $1/\varrho_{max}$ and minimum stretch value $\lambda_{min}$ over the network at each contraction level, for Family-2 $S(\lambda) = \lambda^7 - \lambda^5$ and Family-1 $S(\lambda) = \lambda - 1$ respectively. Note that $1/\varrho_{max} = J_{min}$. Red solid line: cell boundary stretch imposed by boundary conditions. Colorbars: Densification ratio $\varrho$ and fiber radial orientation (in degrees).

at 35–40% contraction, exactly the level of sudden $1/\varrho_{max}$ drop in Fig 4f and band growth in Fig 4e. This shows that element densification is driven by fiber collapse as explained in Fig 2f. We recall also Fig 3m showing a collapsed red fiber within each densified green triangle (see Methods, *Fiber collapse instability*).

The behavior of Family-1 networks is different (Fig 4g, S6 and S7 Figs). The minimal stretch $\lambda_{min}$ follows (12) all the way up to the largest simulated contraction (red line in Fig 4g), occurs on the cell boundary, and is equal to cell boundary contraction prescribed by boundary conditions. This shows that fiber collapse is not observed, as expected. In contrast, the maximal

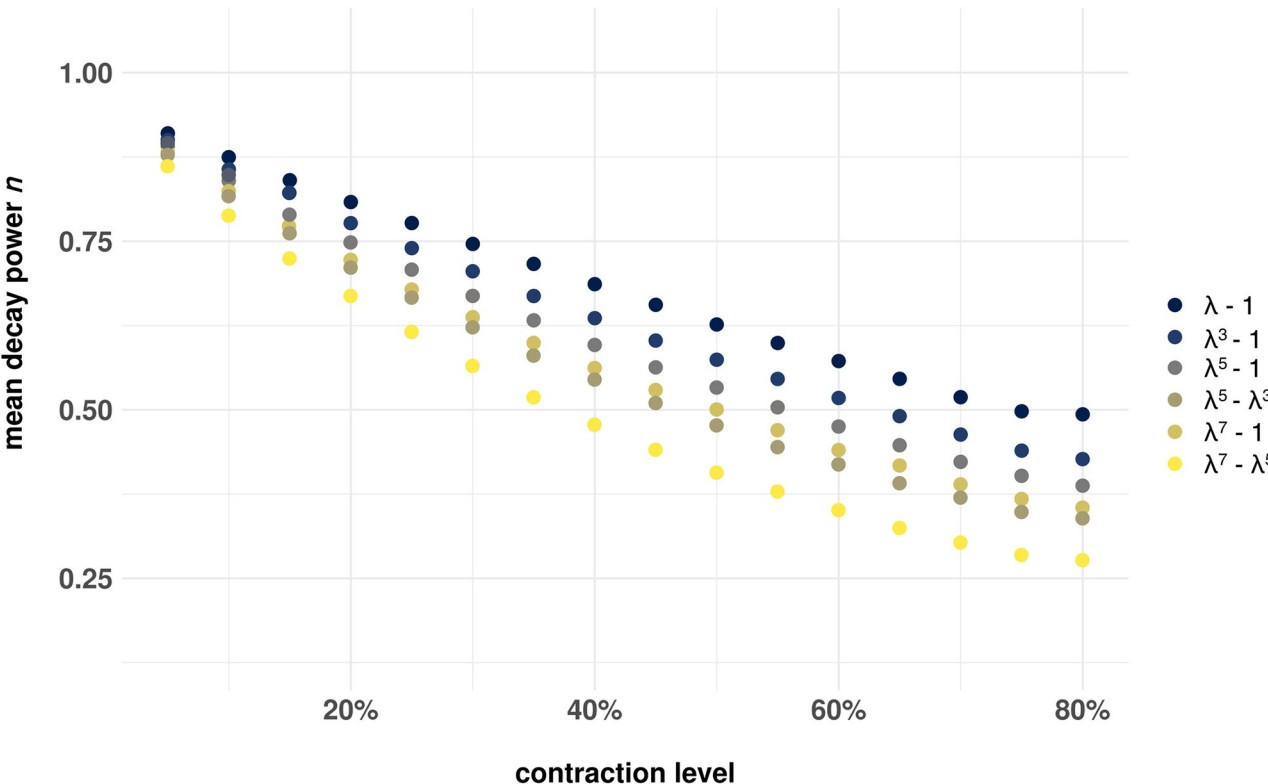

**Fig 5. Mean displacement decay power around a single cell.** Mean displacement decay power $n$ from (13) around a single cell at contraction levels ranging from 5% to 80%. Decay power decreases almost linearly with contraction, at least for contraction levels that reach 50%, for Family-1 models apart from $S\lambda = \lambda^7 - 1$. For the latter along with the Family-2 models there is a monotonic decrease in the decay power. The decay is substantially lower than $n = 1$ (linear elastic solution) even for the linear case.

densification ratio does undergo a sudden leap ($1/\varrho_{max}$ drop in Fig 4g) as in Family-2 models, albeit at a higher contraction level of $\gamma \approx 45\%$–$50\%$. This is evidence of an element collapse instability (see Methods, *Element collapse instability*) that is weaker and requires higher cell contraction than the fiber collapse instability of Family-2 models. Notably, this instability occurs in the linear fiber model $S(\lambda) = \lambda - 1$ (S6 Fig) as well as the nonlinear ones (S7 Fig).

   **Spatial decay of cell-induced ECM displacements.**   A well known behaviour of displacements induced by contracting cells in fibrous ECM is their slower spatial decay rate away from the cell in comparison to predictions of homogeneous linear elasticity, e.g. [18, 32, 34, 36, 37, 40]. Thus, we examine cell contractions within the range of 5% to 80% at different domain sizes, ranging from $5r_c$ to $15r_c$, where $r_c$ denotes the cell radius. We fit the radial displacements of network vertices to the following equation:

$$u(r) \propto A r^{-n} \tag{13}$$

where $u(r)$ represents the radial component of displacement for each node, and $r$ indicates node distance from the center of the contracting cell. The fitting parameters are the constants $A$ and $n$ with $n > 0$ the decay power. Two-dimensional homogeneous linear elasticity, predicts $n = 1$. Therefore, we anticipate $n$ to deviate from the linear elastic solution, providing evidence that our model captures the gradual decay of displacements in natural ECM networks. Our findings are summarized in Fig 5, where the mean decay value is taken over all simulated

domain sizes (disks of radius between 5 and 15 cell radii) at each contraction level. Indeed, we report the decay power $n$ to be substantially lower than the linear elastic value $n = 1$ with strong dependence on cell contraction level, for all our constitutive models, including the linear non-buckling model $S(\lambda) = \lambda - 1$. We find that the decay power ($n$) exhibits a nearly linear decrease with contraction for the linear model at contraction levels roughly $\leq 50\%$. This linearity is somewhat disrupted for family-1 models $S(\lambda) = \lambda^k - 1$, with $k = 3, 5$. However, for models $\lambda^7 - 1$, $\lambda^5 - \lambda^3$ and $\lambda^7 - \lambda^5$ the linearity is lost, replaced by a convex, monotonic decrease in decay power. This behavior remains consistent across varying domain sizes. At higher contraction levels ($> 50\%$), the monotonic decrease is replaced by almost a plateau in decay power $n$ is observed with only slight deviations. For small contractions these results are in qualitative agreement with [40], and the monotonic decrease is also observed in [18], but the linearity is not observed. From our point of view it is not surprising that the linear model exhibits the observed strong dependence on contraction level, since the network behavior is still nonlinear due to large rotations of fibers. The near plateau for very large contractions is very likely due to extreme fiber compaction. This is caused by element collapse instability, which would activate the penalty term in the energy. Another prediction is that the weaker the fibers are in compression, the steeper the decrease of decay power with cell contraction. This explains the steeper decrease of $n$ for smaller bending stiffness observed in [18].

## Intercellular tether formation in two-cell simulations

We report on simulations involving a pair of cells contracting at 50% of their initial radius, separated by either $6r_c$ or $4r_c$, where $r_c$ is the cell radius (Fig 6, S8 and S9 Figs). What distinguishes these from singe-cell simulations is the spontaneous appearance of intercellular tethers, composed of thin, roughly parallel bands of high densification and fiber alignment, that connect the two cells (Fig 6c and S5c Fig). When cells are separated by a larger distance, $6r_c$, tethers are generated only with Family-2 models. Additional densified bands emanate radially from each cell (Fig 6c and S8c Fig) as before. In contrast, in Family-1 simulations, matrix densification is limited close to the cell boundary and cells remain isolated and disconnected (Fig 6a and 6b and S8a and S8b Fig). When cells are closer together, tethers are generated by all models, even the linear one (S9 Fig). In this case, we observe that they are substantially stronger in Family-2 simulations, as they extend from one cell to the other and are noticeably wider compared to Family-1 tethers (S9 Fig, panels (a-c)).

When tethers form, we observe a fraction of fibers, located almost entirely in the intercellular region, to be highly stretched (S9 Fig, panels (d-f)). These fibers are densely packed and aligned with the horizontal direction passing through the cell centers, generating straight paths of fibers connecting the two cells. These paths comprise the tether. At the same time, fibers under extreme compression occupy the same region as the tensile ones, but their orientation is nearly perpendicular to the paths of the tensed and aligned fibers (S9 Fig, panels (g-k)). This is true for all models, though fiber compression magnitude is almost twice as large with Family 2, reaching approximately 70% compression (S9 Fig, panel (k)). This indicates that in Family-2 tethers, compressed fibers are well within the regime of the fiber collapse instability. In addition, we observe highly compressed fibers within the densified bands that emanate radially from the cell periphery (S9 Fig, panel (k)).

When cells are separated by a greater distance, $6r_c$, and tethers are generated only with Family-2 models, fiber stretches highlight a significant difference between families (Fig 6d–6k and S8 Fig, panels (d-k)). In Family-2 simulations, fiber distributions and orientations within the tether are the same as for shorter distances (Fig 6f and 6m, S8f and S8k Fig). For Family-1 models, this is no longer true, as the fiber paths are disrupted and tensile stretches are

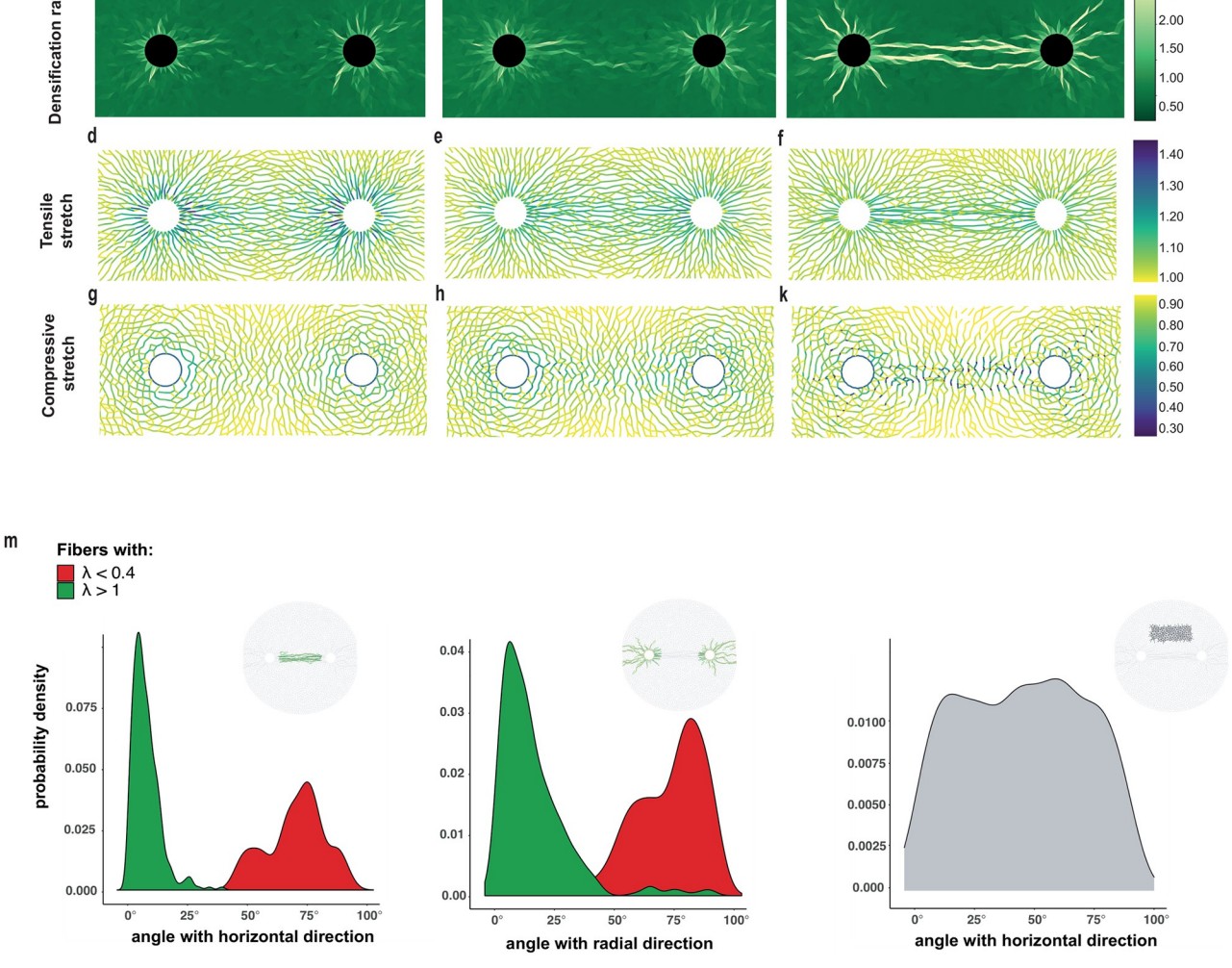

**Fig 6. Intercellular tether formation.** Simulations with two cells contracting at 50%, for three different models (three columns in **a-k**). Cell centers are separated by $6r_c$, where $r_c$ is the undeformed cell radius. **(a-c)** densification ratio of triangular elements (color plot) in deformed networks **(d-f)** tensile stretches and **(g-k)** compressive stretches of deformed fibers **(m)** orientation distribution of deformed fibers within the densified zones (tether and radial bands) in Family-2 case (c) and within the highlighted non-densified zones. Horizontal direction is the one parallel to the axis connecting the cell centres. Radial direction is the one passing through the cell centre. Colorbars: (a-c) densification ratio $\varrho$ of the deformed networks, (d-k) fiber stretch.

distributed in a broader region between cells, without the strong alignment we have with Family-2 models (Fig 6d and 6e and S8d and S8e Fig). This is reflected in angle distributions of the tensile fiber orientation, which are substantially different across models within the intercellular region (S8 Fig, panel (m)). This distribution is more localised for Family-2 models, consistent with greater alignment.

Excessively tensed fiber angles within the densified region are narrowly distributed about zero (horizontal direction through cell centers) (Fig 6m). Compressed fibers are distributed about 80–90° within the densified region, compared to a uniform distribution in non-densified regions (Fig 6m). On the contrary, compressive stretches in Family-1 models are confined to concentric loops around each individual cell instead of the region between cells, and oriented in the circumferential direction (Fig 6g and 6h and S8g and S8h Fig).

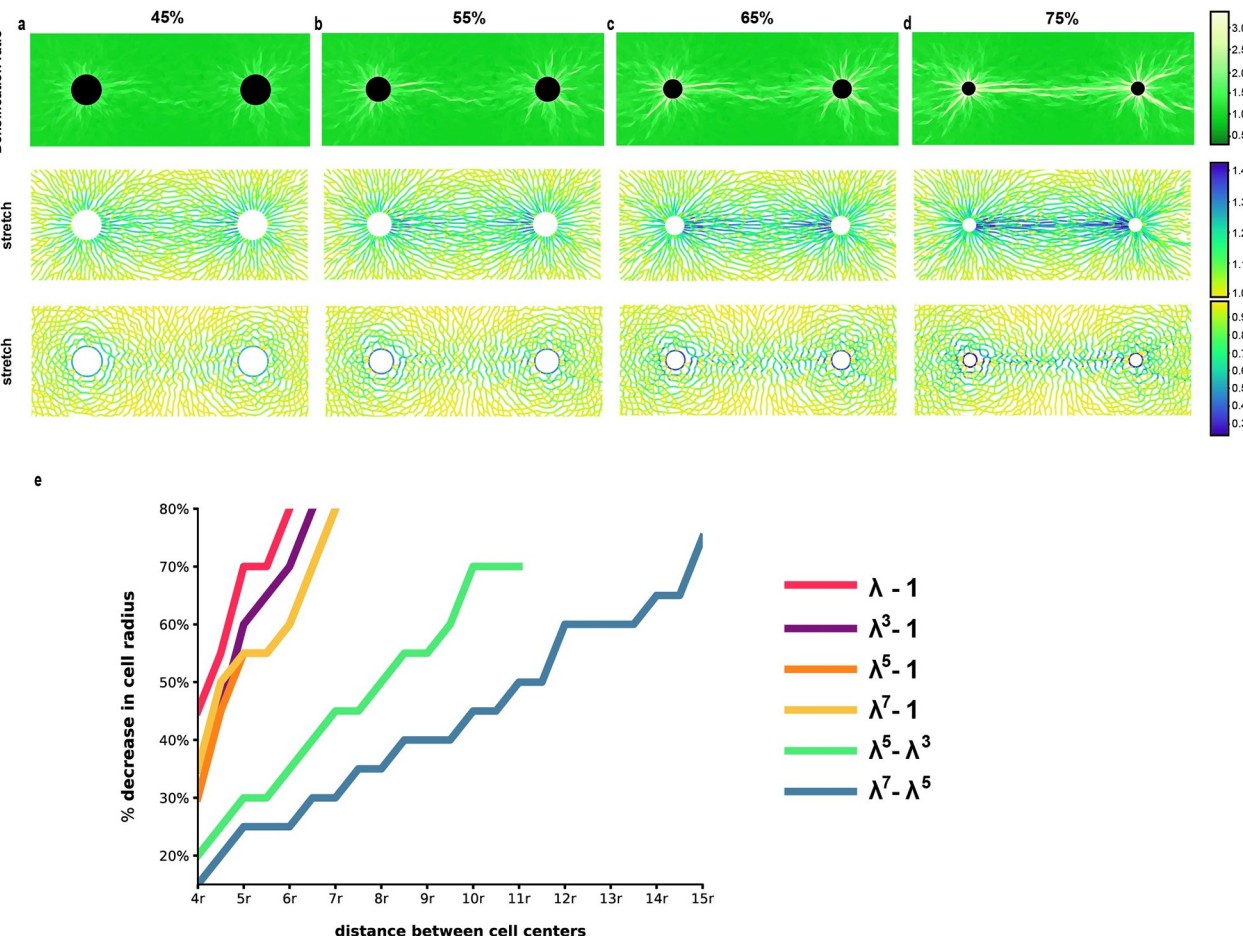

**Fig 7. Tether formation in Family-1 networks. Minimal distance for tether formation for both Family models** Simulations with two cells contracting at different levels in a Family-1 network (model $S(\lambda) = \lambda^5 - 1$). Cell centers have distance $6r_c$, where $r_c$ is the undeformed cell radius. **(a-d)** Densification ratio color plot of triangular elements (up), tensile (middle) and compressive (bottom) stretches of deformed fibers at each contraction step. **(e) Contraction versus cell-cell distance required for tether formation in various models**. Simulations with two cells contracting in the range 5%–80% decrease in cell radius (*y axis*). Cells are separated by a distance proportional to cell undeformed radius *r* (*x axis*). Each curve corresponds to a different model and depicts the minimum contraction level required for formation of a solid tether joining the cells, as a function of distance separating them.

The previous findings hold for 50% contraction. When cells contract more, tethers are eventually generated for Family-1 models as well. In Fig 7 we present the case of Family-1 model $S(\lambda) = \lambda^5 - 1$ (Eq (3), Fig 1c) with two cells separated by $6r_c$ at four contraction levels 45%, 55%, 65% and 75%. We observe that densification between the two cells progressively strengthens. Fiber compression in the cell-cell vicinity is ever-increasing with contraction level, resulting finally in a tether that extends continuously from one cell to the other at 75% contraction (Fig 7d).

Working in the same manner for each model separately, we have tested different contraction levels ranging from 5% to 80% decrease in cell radii, for multiple distances separating the two cells. Results are summarized in Fig 7e. In particular, for each model we obtain a curve that indicates the minimum contraction cells should undergo to produce a tether, expressed as a function of cell distance. That is, above each curve a tether is predicted to form for the respective model. Clearly, Family-2 models are able to sustain tether formation for moderate

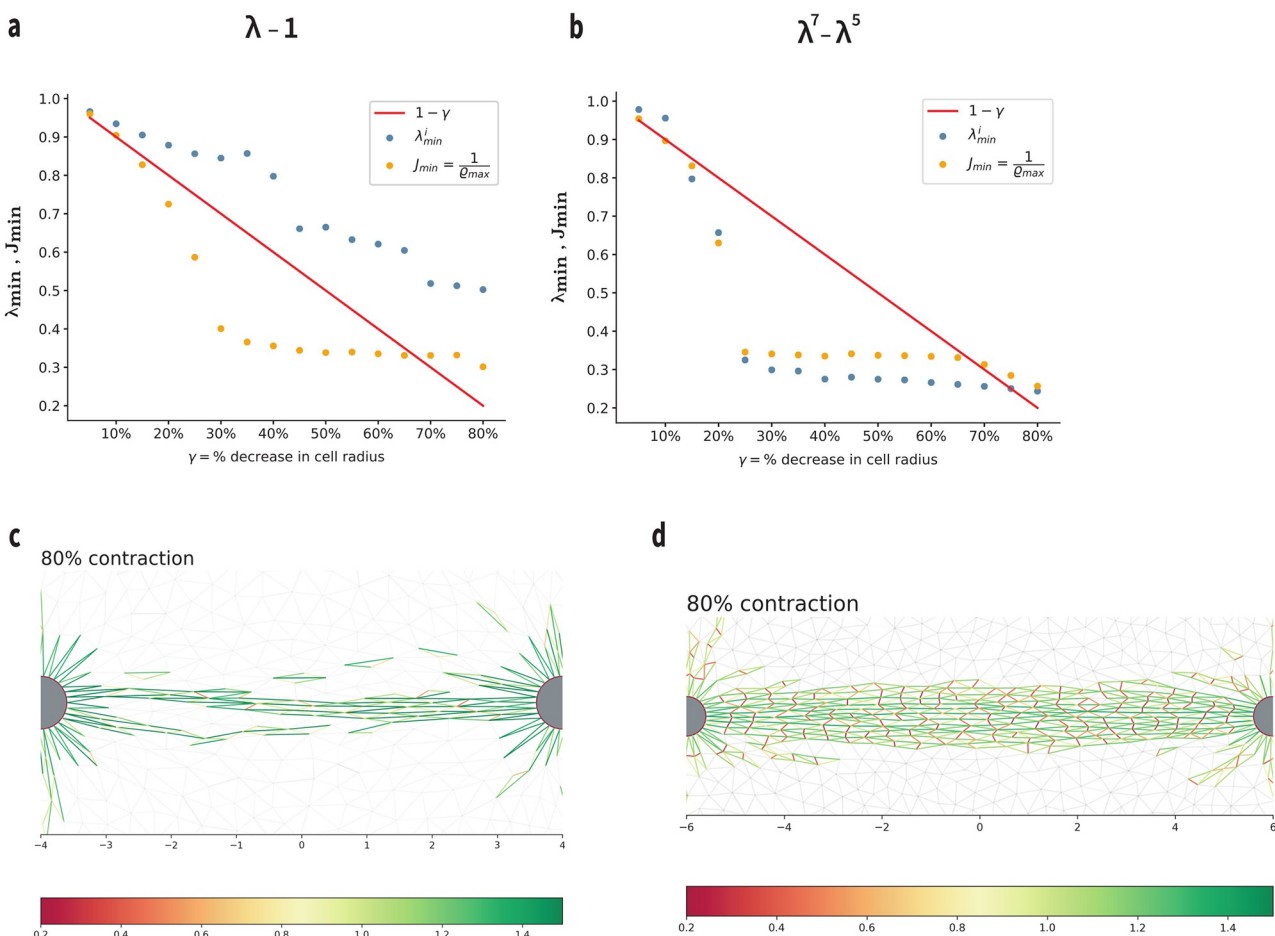

**Fig 8. Mechanisms of densification within tethers in the two Families. (a-b)** Maximum densification ratio inverse $1/\varrho_{max}$ and minimum stretch value $\lambda^i_{min}$ over the network at each contraction level excluding fibers on the cell boundary, for Family-1: $S(\lambda) = \lambda - 1$ and Family-2: $S(\lambda) = \lambda^7 - \lambda^5$ respectively. Note that $1/\varrho_{max} = J_{min}$. Red solid line: cell boundary stretch imposed by boundary conditions. **(a)** Element area collapse (yellow dots) occurs without fiber collapse (blue dots), indicating element collapse instability. **(b)** Densification (yellow dots) occurs simultaneously with fiber collapse (blue dots), suggesting fiber collapse instability. **(c-d)** Stretch of fibers (red: compression; green: tension) at 80% contraction within a Family-1 tether (c) and Family-2 tether (d). Note scarcity of compressed fibers despite presence of collapsed triangles in (c), indicating element collapse instability. In contrast, in (d) most collapsed triangles contain a highly compressed fiber, pointing towards fiber collapse instability.

contraction levels $\leq 50\%$, and for relatively large cell-cell distances (up to $11r_c$). On the contrary, regarding Family-1 models for the same contraction levels $\leq 50\%$, a full tether is formed when cells have a distance at most $5r_c$.

What are the mechanisms responsible for the differences between tethers in the two Family models? Extremely compressed fibers occur in Family-2 tethers ($\lambda_{min} \approx 0.3$) but not in Family 1, where $\lambda_{min} \approx 0.6$ (Fig 8a and 8b). In Family 2, fiber collapse (extreme fiber compression, sudden $\lambda_{min}$ drop in Fig 8b) occurs at the same time as extreme densification (sudden $1/\varrho$ drop, Fig 8b). On the contrary, in Family 1, we see extreme densification without fiber collapse (Fig 8a). In Family 2, most collapsed triangles within the tether contain a highly compressed fiber, oriented within 45° of the vertical (Fig 8d as in Fig 2e). In contrast, in Family-1, collapsed triangles have nearly horizontal bases, while the other two sides are under moderate compression, and are closer to horizontal than vertical after collapse (Fig 8c as in g). These findings

indicate that fiber collapse instability is the main player in Family-2 tethers, whereas the dominant role in Family-1 tethers is played by element (triangle) collapse instability.

**Tethers in lower-connectivity networks.** Most ECM fibrous networks have connectivity that is between 3 and 4 as opposed to 5–6 in our networks [60]. By randomly removing fibers we obtained lower connectivity networks and simulated tether formation for the linear model (Family 1) and the $\lambda^7 - \lambda^5$ Family 2 model. We find that tethers are always weaker in lower connectivity networks, as chains of aligned fibers can be disrupted by missing fibers, but in Family-2 networks tethers remain relatively strong compared with Family-1 tethers, see S10 Fig.

## Discussion

Extracellular Matrix (ECM) mechanical remodeling by cellular forces brings about unique deformation patterns of excessive matrix densification and fiber alignment, both playing a central role in intercellular communication [12, 13, 15, 18] and in cell motility and invasion [8, 21]. In order to explore this type of cell-induced ECM deformations, we develop a discrete model that accounts for individual fibers and their intrinsic mechanics. The discrete fiber network model featured in this study complements our prior work [19], where we demonstrated through continuum modeling and experiments that the aforementioned phenomena are the result of a material instability. The latter is caused by a special nonlinearity due to fiber buckling under compression. To understand how discreteness affects these results, here we implement two different families of fiber constitutive relations, with distinct nonlinearity and stability features. Family 1 displays a positive but decreasing stiffness with increasing compression, while Family 2 entails a stretch instability phase, where stiffness becomes negative at extreme compression. Family 1 represents the traditional view of stable or supercritical [48] post-buckling behavior. Family 2 is a more radical model, incorporating experimental and theoretical evidence for unstable (subcritical) buckling of hierarchical beams, including collagen fibers [47–54]. See Methods, *Evidence for Family 2*.

Our simulation results have revealed instability mechanisms that have not been identified in previous biomechanics work. In particular, all spring networks are susceptible to larger-scale triangular fiber-element collapse instability; see Methods, *Nonconvexity and Instability*. This includes linear-spring networks, provided large rotations are accounted for [41, 56]. In Family-2 networks, an additional fiber collapse instability occurs because of unstable buckling of individual fibers in compression. Simple geometry shows that fiber collapse implies element collapse (Fig 2f), but not vice versa. Thus both instabilities can occur in Family-2 networks. In contrast, Family-1 networks can undergo element collapse, despite the fact that individual fibers exhibit stable behavior. The distinction between these different instabilities is an effect of discreteness and is not captured by continuum models, including ones that involve instability [19]. The present work differs from previous studies in being the first to focus on the role of instability in biological fibrous networks, also to identify, and distinguish between, different instability mechanisms.

In Family-2 models we observe a sudden increase in densification simultaneously with abrupt fiber collapse, both in single- and two-cell simulations (Figs 4f and 8b). The majority of the elements in the densified regions contains a severely compressed fiber (red fibers in Fig 8d). This is strong evidence that the mechanism behind densification in Family-2 networks is fiber collapse instability (Methods and Fig 2f), driven by cell compression. Densification also occurs in Family-1 networks, but requires higher levels of cell compression. Fiber collapse is not encountered in this case (Figs 4g and 8a). Instead, densification is due to triangular elements collapsing (Fig 8c). Element collapse instability (Methods and Fig 2g) is the main player

behind the appearance of densified zones in Family-1 networks. These observations apply to both intercellular tethers and densified zones around single cells.

One important finding concerns fiber alignment. Regardless of the model, simulations show that excessive fiber alignment occurs simultaneously with densification and at the same locations in the ECM, both in single- and two-cell cases (Figs 3 and 6). In particular, in Family 1, we see that severe element compression forces all sides of the densified triangles to align with each other (Fig 8c), a direct effect of element collapse instability (Fig 2g). Especially in two-cell cases, before tethers form, we observe a moderate tendency of fibers to align (Fig 7a and 7b). After tethers form, fibers within the tethers are aligned almost perfectly (Fig 7d) and the elements they belong to are the ones that show extreme densification. In Family-2 tethers and bands, we see the stretched sides of the densified elements aligned with each other, while the highly compressed side is roughly perpendicular to them (Figs 3m and 8d), evidence that fiber collapse instability brings about the alignment of stretched fibers within the densified regions. These significant observations indicate that ECM compression instabilities are responsible for both matrix densification and fiber alignment.

In general it is much easier to form a tether with Family-2 models than with Family-1. For the same distance between two cells, a Family-1 tether requires a much higher compression in order to form. For example, for a distance $6r_c$, where $r_c$ is cell radius, the linear model requires 80% compression whereas 25% is sufficient with $\lambda^7 - \lambda^5$ (Fig 7e). Given a level of compression, say 50%, a Family-1 tether is formed when cells are very close, less than $5r_c$. On the contrary, a Family-2 tether can form when cells are more than twice as far away from each other (Fig 7e).

Perhaps the most prominent prediction of the unstable Family-2 model compared to other models, is the formation of prominently defined, highly localized, strongly fiber-aligned, densified tethers between two cells, and densified radial bands emanating from each cell. Zones of high densification (tethers) have been observed experimentally joining two clusters of contractile cells [12] [13] [17] [61] while thinner bands were seen to emerge from each cluster [12] [13], extending and gradually diminishing within the matrix. Densified tethers and radial bands were also observed in [19] where contracting active particles were used in place of living cells, thereby excluding non-mechanical causes behind densification. Fibers within the tethers are highly aligned along the tether axis. Individual cells from each explant [12] [13] or acinus [17] start migrating along the tether in an attempt to reach the neighboring cluster. Moreover, isolated fibroblasts grew protrusions towards each other, along the tether that formed following their contraction [18]. These studies illustrate the significance of tethers and radial bands in cell migration, motility and intercellular communication. Our simulations identify the formation of these densified zones as a direct consequence of compression instabilities. An additional effect of these instabilities is the close alignment of fibers within the densified zones. Alignment and densification are therefore seen to be part of the same mechanism.

The most essential application of this work is cancer invasion and metastasis. Tumor explants cultured in an initially randomly organized matrix aligned the collagen fibers around them by contracting. This allowed individual cancer cells to use the tracts of aligned fibers as highway paths to invade the ECM [7] [8]. In fact, both densification [62] and fiber alignment [7] are considered prognostic biomarkers for breast carcinoma [63], specifically, "bundles of straightened and aligned collagen fibers that are oriented perpendicular to the tumor boundary" [63]. Contractility is a necessary ingredient in their formation [8]. All these observations apply to both tethers and radial densified bands. A different phenomenon is seen in expanding tumors where the densified layer surrounding them consists of fibers parallel (not perpendicular) to their boundary [63]. Remarkably, our simulations of an expanding cell confirm this, see Supporting Information, S11 Fig. Additionally, aligned collagen fibers provide roads for biochemical molecule transportation between cells [64]. Focusing on the elevated fiber alignment

within the tethers, our predictions highlight the contribution of compression instability to cancer related ECM mechanisms.

Family-2 models (with unstable stretch response) generate well-defined tethers with highly localised densification and fiber alignment very close to the tether axis. These results are in qualitative agreement with experiments [13, 17, 19]. On the contrary, in models with stable stretch response (as in Family 1) tethers are diffused and not localized, while fibers under tension distribute in the broader intercellular region [18, 33–40]. In addition, tethers with Family 2 occur under experimentally observed physiological levels of cell contraction ≈ 50% [17–19], in contrast to Family 1 where extreme contraction is required. Considering the above, Family-2 models are preferable over Family-1 as they describe experimental observations better. For evidence in support of the unstable response of Family-2 fibers [47–54], see Methods *Evidence for Family 2 Unstable Fiber Buckling Model*.

There have been considerable efforts in ECM modeling [18, 32–40]. In these studies, ECM fibers are usually modeled as Timoshenko beams [33, 37] or as elements with asymmetric elastic responses to extension and compression, obeying either piecewise linear stress-strain curves or combining strain-stiffening with compression softening intended to model fiber buckling [18, 32, 35, 36, 38–40]. Even though these approaches have explored nonlinear aspects of fiber behavior, they are limited by stable stretch responses (monotonic) and sometimes small deformations. As far as the stretch response is concerned, all these models are similar in spirit to our Family-1 models. None of these works address instability. Here, we recognise that instability plays a central role in the appearance of ECM densification and fiber alignment. Our work shows that instability can occur even in models with stable stretch response (Family 1), but it does so at cell contraction levels that are substantially higher than experimentally observed ones. This casts doubt on the biological relevance of Family 1, and prompts us to introduce models (Family 2) whose stretch response becomes unstable in compression. This particular instability allows tether formation and fiber alignment under experimentally observed cell contraction levels.

Fiber alignment has been explored by previous studies [38], [40], yet as a separate ECM mechanism involved together with compression buckling in intercellular force transmission [38] or matrix elastic anisotropy [40]. Sopher et al. [38] suggest that elevated tension in the intercellular region obliges fibers to stretch and align. In our simulations densification and fiber alignment suddenly jump to much higher values compared to aforementioned works. This transition occurs when a compression instability is triggered (either a fiber or an element collapses). For models with stable stretch response (Family 1), this would occur in much higher contraction levels than considered in previous works [18, 32, 35, 36, 38–40], which explains the substantially lower levels of densification and alignment observed in those works, none of which consider instability mechanisms, while some report absence of buckling [65]. In comparison with these works and Family-1 models, Family-2 models not only require moderate cell contraction, but the resulting densification is much stronger, tethers are much more sharply delineated through discontinuous density localization, and fibers are almost perfectly aligned to the tether axis.

Deformation induced anisotropy has been studied [40] as a possible mechanism for long-range cell communication. We remark that the compressive instabilities studied here create strong anisotropy in the densified state of the network, because they create a strongly aligned and dense uniaxial distribution of fibers from an originally roughly isotropic random fiber distribution. Our results show that compression instability provides a single unifying mechanism that accounts for all three phenomena of densification, fiber alignment and matrix anisotropy. These phenomena occur simultaneously within the same localized zones as soon as

compression instability is triggered either due to fiber buckling (collapse) or element collapse (snap-through buckling).

Compression instability due to buckling of elastic fibers in networks was first identified in [44] as a mechanism of localized densification. The continuum model of [19, 42], obtained from a fiber model through orientational averaging, also exhibits a compression instability, and predicts highly localized densified tethers and radial bands. Their morphology is qualitatively similar to the ones reported here. In general, it seems practically impossible to obtain an explicit continuum constitutive law for a random network through a rigorous limit as the fiber length approaches zero [41, 66]. The advantage of our model is that it captures the discrete nature of the actual fibrous network. As a result it can distinguish between different types of compression instabilities (buckling of fibers versus fiber elements), some of which are only possible in discrete network type media and not in uniform continua, and it clarifies the close connection of instability with fiber alignment and densification.

Apart from buckling, other explanations have been proposed for inhomogeneous ECM deformations, such as filopodial growth, [39, 64, 67], and heterogeneous or low connectivity networks [65, 68]. In experiments (see Fig 5 in [18]) with two fibroblasts, a single densified tract joining them appears immediately upon cell contraction, and subsequently, filopodia gradually extend along the densified tract. In [19], experiments with spherical, actively contracting particles in place of cells in the ECM, revealed immediate appearance of tethers and densified radial hairlike bands emanating from the contracted particles, which maintain sphericity upon contraction, and cannot produce protrusions. This shows that filopodia are not necessary for densification pattern formation. Low connectivity networks and other networks with strongly inhomogeneous geometries will automatically produce inhomogeneities in the equilibrium displacements. This is not surprising, as inhomogeneity in the forces (caused by filopodia) and in the network geometry are bound to cause inhomogeneity in the deformation. For example in [33] circular cells form tethers only at huge contration levels (as in Family 1 here) whereas pointed narrow ellipses cause inhomogeneity that triggers tethers at lower contraction levels due to stress concentration at their tips. Here instability causes much more severe inhomogeneous deformations in initially perfectly homogeneous continua [19] or in nearly homogeneous networks in the present work. We find here that the highly unstable Family 2 produced much stronger and discernible tethers than Family 1, see S10 Fig, whereas low connectivity in fact weakens densification bands by breaking aligned fiber chains. The type of inhomogeneity caused by compressive instabilities, is distinguished by localized bands of strong densification, unlike the more random inhomogeneities discussed above.

There are alternative models that do predict well defined, strongly localized tethers with very high alignment, but apart from linear fibers they also propose irreversible crosslink sliding and fiber merger [69], and "agent-based" evolution of crosslinks [70]. We remark that irreversibility due to fiber merger was also modeled in [19] via an additional attractive fiber potential that amplifies some collapse instabilities. It was found however [15, 19] that in fully crosslinked networks, densification appears upon cell contraction and disappears fully upon re-expansion; so plasticity-like irreversible effects are not essential for the formation of densified tracts and tethers.

Possible extensions of our work include three-dimensional network models, more sophisticated modeling of joints and crosslinking between fibers, viscoelastic effects in fiber behavior, and the possibility of larger-scale instabilities. Also, a better understanding of similarities and differences with periodic discrete networks [41] and other periodic media [45, 71] where instabilities occur at multiple scales and are known to play a crucial role.

Our models highlight compression instability due to buckling as a crucial nonlinear mechanism underlying the mechanical behavior of fibrous ECM and give rise to new insights in

exploring the nature of cell-induced deformations that underlie matrix densification and fiber alignment and their implications in intercellular biomechanical interaction, cancer metastasis and cell motility.

## Supporting information

**S1 Fig. Interpenetration of matter and penalty term.** **(I)** Various presentations of triangulated rectangular truss elements. Each edge in the structures represents a linear spring. Dirichlet boundary conditions were applied on the upper boundary nodes by imposing a displacement u = ($h$, 0.0), $h$ being the scale to $x$ direction. Deformed structures contain triangles that have changed orientation, resulting in interpenetration of matter. **(II)** Top: Penalty term $\Phi(J) = exp(-Q(J - b))$, where $J$ is *ratio of deformed to undeformed oriented triangle area*. $Q > 0$ is large and $b > 0$ is small constant. As a result, negative values of $J$ have high energy cost, whereas positive values have negligible contribution to the network's total energy. Bottom: Simulations of a cell contracting by 50%, either with or without the penalty term for the area ratio $J$. Without penalizing $J$, the optimizer finds solutions that are physically unacceptable, as $J < 0$ corresponds to elements (red) that changed orientation. Colorbar: $J$ values.
(TIF)

**S2 Fig. Penalty term—The role of parameter *b*.** The penalty term is given by $\Phi(J) = exp(-Q(J - b))$, where $J$ is *ratio of deformed to undeformed oriented triangle area*. We illustrate simulations of a cell contracting by 50%, employing the penalty term with $Q = 50$ and $b = 1/3, 1/4, 1/6$ or $1/8$. The obtained solutions exhibit qualitative similarity. Similar results are observed when varying $Q$ while keeping $b$ constant. Colorbar: densification ratio of the deformed networks.
(TIF)

**S3 Fig. Fiber collapse instability and severe localized densification.** Complementary to Results, Fig 3 containing simulations of a single cell at 50% contraction with Family-1 models $\lambda^3 - 1$ and $\lambda^7 - 1$ and Family-2 model $\lambda^7 - \lambda^5$. **(a-c)** Densification ratio of triangular elements (color plot) in deformed networks **(d-f)** tensile stretches and **(g-k)** compressive stretches in deformed fibers. **(m)** Minimum contraction required for densification to be evident for each one of the models studied. **(n)** Simulations with various models of one cell contracting at 50%; *x axis:* triangular element distance from cell center, *y axis:* element densification ratio. Colorbars: (a-c) densification ratio $\varrho$ of the deformed networks, (d-k) fiber stretch $\lambda$.
(TIFF)

**S4 Fig. Effect of contractile particle size vs fiber size on radial densified band formation.** (a) A cell or small particle whose radius is a few fiber lengths (fiber network is visible as the mesh) produces very few densified bands with densification ratio above 2, whereas at the same contraction level of 50% a larger particle produces a great number of radial hairs with higher densification ratio exceeding 3. The model in a) and b) is Family 2 $S(\lambda) = \lambda^5 - \lambda^3$. (c) Same mesh (fiber network) as in (b) but with the Family 2 $S(\lambda) = \lambda^7 - \lambda^5$ model. Hairs are substantially longer than in (b). Colorbar: densification ratio of the deformed network.
(TIFF)

**S5 Fig. Comparison of area densification ratio and fiber density.** Simulated area densification (a) and fiber density (b), both obtained from the same simulation of an active particle undergoing 50% contraction. Area densification ratio is much more inhomogeneous than fiber density, and densified tracts appear fewer and further apart in (a) versus (b). The length of densified tracts as defined by area densification (a) is much more objective than when

defined by fiber density (b), because the former exhibits a sharp discontinuity and is largely independent of photographic scales such as exposure, contrast, brightness etc. Also it is more difficult to compare it to available experimental images, such as (c) (adapted with uniform contrast increase from Fig 5b of [19] with permission) which are sensitive to photography and its electronic reproduction at various stages. Here (c) is juxtaposed with a simulation (d) for comparison. Unfortunately, the light intensity versus fiber density curve from experiments reported in [19] is not available, and there is a subjective choice of brightness/contrast scale in both experimental images (c) and simulations (b), (d). However, it is clear that there are radial hair structures in both. Their apparent length depends on brightness vs fiber density scales that are to some extent arbitrary, but the topology and pattern is similar and much more objective. In (c) we have increased the experimental image contrast uniformly across the image, to emphasise the nonuniform fibers emanating from the particle.
(TIFF)

**S6 Fig. Progressive cell contraction and matrix localization.** Complementary to Results, Fig 4. Simulations with the linear Family-1 model $S(\lambda) = \lambda - 1$ of a cell contracting in the range 5%–80%. *Top:* densification ratio $\varrho$ color plot at each indicated contraction step. *Middle:* tree diagrams, fiber distance from cell center versus fiber stretch for all fibers in the network at each contraction step, x axis: fiber stretch, y axis: fiber distance from cell center. *Bottom:* triangular element distance from cell center versus densification area ratio, *x axis:* densification area ratio $\varrho$, *y axis:* triangular element distance from cell center.
(TIF)

**S7 Fig. Progressive cell contraction and densification. (a-b)** As contraction level rises, densification strengthens in the close proximity of the cell for Family-1 models. The bands consisting of densified elements do not propagate far from the cell boundary, reaching as far as 3 deformed cell radii at 80%. On the contrary, in Family-2 simulations **(c-d)** densification is evident at much lower contraction levels, 35%. With increased contraction, more densified bands are generated and extend substantially further into the matrix. Colorbar: densification ratio $\varrho$ of deformed networks.
(TIF)

**S8 Fig. Intercellular tether formation.** Complementary to Results, Fig 5 containing predictions for the remaining models. Simulations with two cells contracting at 50%. Cell centers are separated by $6r_c$, where $r_c$ is the undeformed cell radius. **(a-b)** densification ratio of triangular elements (color plot) in deformed networks **(d-f)** tensile stretches and **(g-k)** compressive stretches of deformed fibers. **(m)** Orientation distribution of fibers under tension (stretch $\lambda > 1$) within the intercellular region across all models. Each violin corresponds to each one of the models studied and shows the distribution of fiber horizontal direction (in degrees), ***$p - value < 0.001$. Colorbars: (a-c) densification ratio of the deformed networks, (d-k) fiber stretch.
(TIFF)

**S9 Fig. Intercellular tether formation when cells are separated by a shorter distance.** Simulations with two cells contracting at 50%. As in S8 Fig except that cell centers are separated by $4r_c$, where $r_c$ is the undeformed cell radius. **(a-b)** densification ratio of triangular elements (color plot) in deformed networks **(d-f)** tensile stretches and **(g-k)** compressive stretches of deformed fibers. We observe densification around each cell boundary, which extends towards the neighboring cell. Tethers are rather weak for Family-1 cases (a-b) and significantly stronger with Family-2 (c). Within tethers, densification ratio is three times larger than the rest of the matrix. In the intercellular region, fibers under tension are directed towards the neighboring

cell so that they form continuous paths connecting the two cells. In these paths, fibers under tension are almost perfectly aligned with the horizontal line connecting the two cells. In Family-2 case (f) excessive tensile stretches are concentrated only within the tether-region. Severely compressed fibers (g-k) locate in the intercellular domain, being roughly perpendicular to fibers under tension. Colorbars: (a-c) densification ratio of the deformed networks, (d-k) fiber stretch.
(TIF)

**S10 Fig. Tether formation in networks with lower connectivity.** Simulations with two cells contracting at 50% with Family-1 model $\lambda - 1$ (a-f) and Family-2 model $\lambda^7 - \lambda^5$ (g-p). Cell centers are separated by $4r_c$, where $r_c$ is the undeformed cell radius. We examined how decreasing the network connectivity ($C$) impacts the alignment of fibers between adjacent cells. We report on simulations with $C \sim 6$ for a fully connected network, $C \sim 4.7$ or $3.6$ for networks with lower connectivity. A direct observation of these simulations is the persistence of fiber alignment with the Family-2 model, even at the lowest connected network ($C \sim 3.6$). Moreover, expanding upon the primary observation in fully connected networks, where fiber alignment coincides with fiber compression, we observe that this phenomenon persists in less connected networks. This holds for both Families. Despite alterations in geometry, our findings indicate that compression instability facilitates fiber alignment. Colorbar: stretch of deformed fibers.
(TIF)

**S11 Fig. Cell expansion.** Simulation with $S(\lambda) = \lambda^5 - \lambda^3$ of a single cell radially expanded by 50%. **(a)** Densification ratio of triangular elements (color plot) in deformed networks **(b)** compressive stretches and **(c)** tensile stretches in deformed fibers. Note that the compressed fibers align with the radial direction while fibers under tension orient in the angular direction.
(TIF)

## Author Contributions

**Conceptualization:** Georgios Grekas, Phoebus Rosakis.

**Data curation:** Chrysovalantou Kalaitzidou.

**Formal analysis:** Chrysovalantou Kalaitzidou.

**Funding acquisition:** Chrysovalantou Kalaitzidou, Andreas Zilian, Charalambos Makridakis.

**Investigation:** Chrysovalantou Kalaitzidou.

**Methodology:** Georgios Grekas, Phoebus Rosakis.

**Resources:** Andreas Zilian, Charalambos Makridakis.

**Software:** Chrysovalantou Kalaitzidou, Georgios Grekas.

**Supervision:** Georgios Grekas, Andreas Zilian, Phoebus Rosakis.

**Visualization:** Chrysovalantou Kalaitzidou.

**Writing – original draft:** Chrysovalantou Kalaitzidou.

**Writing – review & editing:** Phoebus Rosakis.

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
