## [Decision Letter · Decision Letter 0]

22 Aug 2023

Dear Ms Kalaitzidou,

Thank you very much for submitting your manuscript "Compressive Instabilities Enable Cell-Induced Extreme Densification Patterns in the Fibrous Extracellular Matrix: Discrete Model Predictions" for consideration at PLOS Computational Biology.

As with all papers reviewed by the journal, your manuscript was reviewed by members of the editorial board and by several independent reviewers. In light of the reviews (below this email), we would like to invite the resubmission of a significantly-revised version that takes into account the reviewers' comments.

The reviews of your manuscript have been received. All three reviewers provided specific suggestions for improving the manuscript. Several reviewers suggested improvements for more clearly introducing and motivating the present study and also clarifying the assumptions of the model. The reviewers also noted several studies using similar approaches that should be compared and discussed in the context of the present work. Reviews also note that model code is not yet available for their review.

We cannot make any decision about publication until we have seen the revised manuscript and your response to the reviewers' comments. Your revised manuscript is also likely to be sent to reviewers for further evaluation.

Sincerely,

Seth Howard Weinberg, PhD

Guest Editor

PLOS Computational Biology

Mark Alber

Section Editor

PLOS Computational Biology

The reviews of your manuscript have been received. All three reviewers provided specific suggestions for improving the manuscript. Several reviewers suggested improvements for more clearly introducing and motivating the present study and also clarifying the assumptions of the model. The reviewers also noted several studies using similar approaches that should be compared and discussed in the context of the present work. Reviews also note that model code is not yet available for their review.

Reviewer's Responses to Questions

**Comments to the Authors:**

Reviewer #1: In this work, Kalaitzidou et al. developed a nonlinear fiber network model of fibrous collagen ECM. Fibers are modeled as the edges of a triangular mesh and two families of models are simulated for evaluating the role of instabilities in ECM remodeling. They provide simulations that demonstrate spontaneous ECM remodeling due to cell contraction.

Overall, the model and results are interesting. However, there are various concerns that prevent me from recommending the paper for publication in its current form. I have detailed these concerns below.

1. In the abstract, the authors claim, “This has been avoided so far...”. What has been avoided? Understanding the biomechanics of ECM, or fully accounting for nonlinear instability?

2. In the abstract, authors say that they use “cartoon-like models”. Do they mean simplified use-cases?

3. In the abstract (and similarly in the author summary), the authors state “remodelling creates complex patterns, comprising narrow localised bands of severe densification and fiber alignment, extending unusually far into the ECM...”. It is unclear what the authors describe as unusual, whether the extent of densification is unusual as observed in experimental observations or in the predictions of their model.

4. In the abstract, the authors describe their model as “fully nonlinear”. It is unclear what “fully” vs. “partially” nonlinear means and what features of the model make this claim true or defensible.

5. The content of the abstract is not well organized. For example, the authors refer to justifications of their modeling assumptions without describing their assumptions. They also jump around a lot between what is observed, what others have done, and what they did in this work.

6. In the abstract, the authors claim that no other model “provides a simple mechanism of compressive instabilities that fully explains the phenomenon”. It’s unclear what they mean by “fully explain”. Presumably they don’t mean that their model explains everything involved in densification, fiber alignment and emergent patterns.

7. In the abstract, the authors claim that their model is “simple and easy to comprehend”. This claim as written entirely depends on the knowledge and expertise of the reader and so I don’t think that it is an appropriate claim to make without some justification or clarification.

8. In the abstract (and similarly throughout), the authors describe “striking patterns of ECM pathways” and claim that their model predicts them but provide no quantitative metrics to judge how well such predicted patterns match those observed in any experiment or justification as to why qualitative assessment is sufficient. This diminishes the reproducibility and credibility of the work, and the authors should address this by either providing such metrics or, in the absence of any established metrics to do so, should state as such.

9. Similarly, in line 413 the authors claim that a solid tether forms and then further develop with this concept in the following paragraph (e.g., Figure 6). It’s unclear what constitutes a “solid tether” vs. “not a solid tether” besides a qualitative assessment, which also diminishes the reproducibility and credibility of the work. Later (line 497) the authors somewhat define tethers, though qualitatively, as “zones of high densification”.

10. In the author summary (lines 5-6), the authors claim that “Observed response of the ECM to forces from contracting cells is unexpected”. By response, do they mean mechanical or chemical? One would obviously expect a mechanical response by fibrous ECM to externally applied forces, but also previous models of other ECM components have demonstrated responses like fibronectin fibrillogenesis (https://doi.org/10.1038/s41598-017-18328-4) to externally applied forces.

11. In the Author Summary (lines 10-12), the authors state that they "hypothesise that this behaviour is caused by the inability of fibers to resist compression, much like rubber bands, and the fact that in a network, there is room for fibers to rotate, buckle and collapse”. Which behavior(s) is the authors hypothesizing is due to these fiber characteristics?

12. Figure 7 subpanels a and b need labels on their vertical axis.

13. In line 559, the authors refer to a compression instability being “activated” but it is not unclear what exactly is an “activated compression instability”.

14. In line 570, the authors claim that their “results show that there is a single unifying mechanism behind densification, fiber alignment and matrix anisotropy, …". Certainly, they have shown that this is true for their model. But the wording reads as if the authors are claiming that their results show that this is how we should understand the mechanisms responsible for densification, fiber alignment and matrix anisotropy. Either this language should be refined to narrow the scope of the claim, or some justification should be made as to why other mechanical and chemical features of the ECM neglected by their model are unimportant to our understanding of these phenomena.

15. The authors describe in line 550 that “unreasonably high cell contraction levels” were simulated and reported in the work and refer to “experimentally observed cell contraction levels” in lines 552-553. It’s unclear how to interpret simulation results, whether as biologically or only theoretically meaningful.

16. Given that simulations produced fibers that nearly collapsed (e.g., line 304), the sensitivity of results to the parameters “Q” and “b” in Equation 8 seems important. However, the authors do not show whether either parameter has any effect on results.

Reviewer #2: Understanding how cells mechanically interact with the extracellular matrix is a hot topic that has relevance for many fields in biology. In this manuscript, the authors develop a discrete fiber network model to describe extracellular matrix mechanics, and investigate how a circular inclusion mimicking a contracting cell deforms the matrix. They construct a network made of triangular elements, and compare two different families of constitutive models for the force-extension behavior of individual fibers (sides of the triangles). Both families of models account for stiffening in tension and softening in compression, while family 2 additionally accounts for non-linear behavior ("stretch-instability") in compression.

The manuscript is very well written, with clear and polished language. The results are interesting and explained well. The authors give a good broad introduction and discuss large parts of the relevant literature.

Some of the claims are stated too broadly, and some points need to be clarified further, which the authors can address with a minor revision.

%%%% Major comments

1) The authors claim that their previous continuum model (reference 18) and their new discrete model in this manuscript predicts thin hair-like densification which they describe as "localized densification patterns that bear a strong resemblance with experiments [13, 16, 18]." (lines 56-57). However, the cited literature shows these patterns only in multicellular aggregates, where it is unclear if they emerge purely due to mechanics or due to cell migration or extension of filopodia into the tethers. Experimental work with single rounded cells does not seem to have these extremely localized heterogeneities, unless filopodial protrusions are included (see e.g. Mann et al. 2019 [reference 38 in the manuscript], Malandrino et al. 2019, Gomez et al. 2019 [reference 56 in the manuscript]). Some of the authors' previous experimental work using synthetic contractile particles also lacks the thin highly heterogeneous densification bands (Grekas et al. 2021 [reference 18]). This would suggest that their family 1 model might be a better suited for single cell force-transmission, while their family 2 model might better represent matrix deformation by multicellular aggregates.

The authors should revise the statements in the introduction and extensively discuss this point.

Malandrino, Andrea, et al. "Dynamic filopodial forces induce accumulation, damage, and plastic remodeling of 3D extracellular matrices." PLoS computational biology 15.4 (2019): e1006684.

2) One central claim is that the authors' model is unique in capturing long-ranged and spatially heterogeneous densified bands of extracellular matrix around a contractile inclusion. It is true that most discrete fiber network models have very homogeneous deformations around the inclusion, but there are examples of heterogeneous deformation, e.g. for some network geometries (Humphries et al. 2017), and for networks with low connectivity (Slater et al. 2021, Ruiz-Franco and van der Gucht 2021, Tsingos et al. 2023). This previous work should be acknowledged and discussed. It would be interesting to highlight qualitative differences between the densification heterogeneity observed by the authors and by previous work.

Humphries, D. L., J. A. Grogan, and E. A. Gaffney. "Mechanical cell–cell communication in fibrous networks: the importance of network geometry." Bulletin of mathematical biology 79 (2017): 498-524.

Ruiz-Franco, José, and Jasper van Der Gucht. "Force transmission in disordered fibre networks." Frontiers in Cell and Developmental Biology 10 (2022): 931776.

Slater, Brandon, et al. "Transient mechanical interactions between cells and viscoelastic extracellular matrix." Soft Matter 17.45 (2021): 10274-10285.

Tsingos, Erika, et al. "Hybrid cellular Potts and bead-spring modeling of cells in fibrous extracellular matrix." Biophysical Journal (2023).

3) Related to point 2. Fiber densification: The authors define fiber densification by taking the ratio of the area of each triangular element to its undeformed reference area. In experiments, typically the density of fibers per unit area is quantified. The authors should verify (numerically) that these two metrics are indeed directly comparable.

4) Related to point 2. What is the average connectivity of the networks the authors studied? For reference, average connectivity in collagen matrices is between 3 and 4 (Jansen et al. 2018, Burla et al. 2020).

Burla, Federica, et al. "Connectivity and plasticity determine collagen network fracture." Proceedings of the National Academy of Sciences 117.15 (2020): 8326-8334.

Jansen, Karin A., et al. "The role of network architecture in collagen mechanics." Biophysical journal 114.11 (2018): 2665-2678.

5) Line 547: The authors claim that previous discrete network models of the matrix do not address instability. If the authors refer to buckling, then the statement is wrong, since others have investigated the effect of buckling, e.g. Notbohm et al. 2015 (cited as reference 17). The authors should clarify what they mean by instability and revise the statement accordingly.

6) It is not clear what the authors mean by "rotations". Are these rotations of individual fibers, of triangles, or of the order of vertices in a triangle (clockwise vs counterclockwise)? Related to this, the authors claim that previous models are restricted to small rotations, but it is unclear what is meant. Please explain.

%%%% Minor comments

It would be interesting for the authors to quantify the force transmission distance, as this would allow them to compare their data to previous work. See section 4.3 of the following paper for an explanation.

Wang, Haiqin, and Xinpeng Xu. "Continuum elastic models for force transmission in biopolymer gels." Soft Matter 16.48 (2020): 10781-10808.

Line 201: Typo, "neighoring" should be "neighboring"

Line 206: Unexpected newline

Line 207: Sentence should end with a full stop.

Line 288: "continual" should be "continuous"

Line 453: Typo, "Our simulations results" should be "Our simulation results"

Line 524:

---

## [Decision Letter · Decision Letter 1]

26 Feb 2024

Dear Ms Kalaitzidou,

Thank you very much for submitting your manuscript "Compressive Instabilities Enable Cell-Induced Extreme Densification Patterns in the Fibrous Extracellular Matrix: Discrete Model Predictions" for consideration at PLOS Computational Biology.

As with all papers reviewed by the journal, your manuscript was reviewed by members of the editorial board and by several independent reviewers. In light of the reviews (below this email), we would like to invite the resubmission of a significantly-revised version that takes into account the reviewers' comments.

We cannot make any decision about publication until we have seen the revised manuscript and your response to the reviewers' comments. Your revised manuscript is also likely to be sent to reviewers for further evaluation.

Sincerely,

Seth Howard Weinberg, PhD

Guest Editor

PLOS Computational Biology

Mark Alber

Section Editor

PLOS Computational Biology

Reviewer's Responses to Questions

**Comments to the Authors:**

Reviewer #1: The authors have made significant improvements to their manuscript and addressed most of my concerns.

In their response to reviewers, the authors mention a “likely difference of culture between authors and readership of the Journal”. I’m confused about the intent of this statement given that scientists from engineering, medical, and mathematics departments (like the authors) commonly publish in (and so presumably read) PLOS Computational Biology, and about how this statement is relevant to my feedback. I bring this up because of my concerns that were not addressed in revisions, specifically on the organization of content in the abstract and reproducibility of their results. I assume that these concerns were not addressed for other reasons and not because of presumed culture differences but, in case my assumption is incorrect, would like to point out that neither concern is particular to disciplinary background or culture. They are issues of the rigor and potential impact of the work.

Comment #5: Revisions address detailed concerns in the abstract but do not try to improve the organization of the content. At various times, the abstract jumps back and forth between the innovation, assumptions, and relevance of the work in this project, limitations in other models, motivating biology, etc., in a haphazard way that makes it difficult (at least to me) to perceive the overarching story of the work. I’m disappointed that the authors were not responsive to this feedback, which was intended to improve the likelihood of attracting readers to their paper.

Comment #8: The issue I raised in this feedback was more concerned with addressing the reproducibility of the work in general. Qualitative prediction is subjective, and if that’s the best that can currently be done, then so be it. I am asking the authors to address the issue per se. Reproducibility (or the lack thereof) in biology is a rapidly growing concern, to which biological modeling is not immune. How can someone else implement the model as described in the manuscript and assess whether they have done so correctly? How could predictions from the model be quantitatively compared to experimental results? If the answer to such questions is “they can’t”, then why not? Is there no well-defined technique using image analysis, morphometrics, etc., to address these limitations?

Reviewer #2: The review is uploaded as an attachment.

Reviewer #3: In my previous review, I showed a concern about lack of evidence for the snap-through collapse of ECM fibers and the dependence of the ECM model on the unverified assumption. In their responses, they argued that an assumption without direct evidence can be meaningful if it enables the model to reproduce experimental observations, and that there is indeed indirect evidence for their assumption. Well, I have to agree to disagree with them. There have been a few studies showing the buckling of collagen fibers with the formation of waves or loops: DOI: 10.1016/j.actbio.2022.06.044 and 10.1021/acsnano.0c03695. I don't think that the observation of these patterns during buckling supports the snap-through collapse of collagen fibers. They look like the buckling behavior of linear beam structures. The indirect evidence that the authors referred to seems to be theoretical prediction or different structures. After seeing these experimental observations directly related to collagen fibers, it is hard to buy the authors' assumption even more. If the assumption is not verified, I would consider a model based on the assumption to be a phenomenological model, which shows one of many possible ways to reproduce experimental observations at the ECM scale. Then, the impact of such a model would be limited.

**Have the authors made all data and (if applicable) computational code underlying the findings in their manuscript fully available?**

Reviewer #1: **No: **

Reviewer #2: Yes

Reviewer #3: Yes

PLOS authors have the option to publish the peer review history of their article (what does this mean?). If published, this will include your full peer review and any attached files.

Reviewer #1: No

Reviewer #2: No

Reviewer #3: No
---

## [Decision Letter · Decision Letter 2]

8 Jun 2024

Dear Ms Kalaitzidou,

We are pleased to inform you that your manuscript 'Compressive Instabilities Enable Cell-Induced Extreme Densification Patterns in the Fibrous Extracellular Matrix: Discrete Model Predictions' has been provisionally accepted for publication in PLOS Computational Biology.

Best regards,

Alison Marsden

Section Editor

PLOS Computational Biology

Alison Marsden

Section Editor

PLOS Computational Biology

Reviewer's Responses to Questions

**Comments to the Authors:**

Reviewer #1: The authors have sufficiently addressed all of my concerns.

Reviewer #2: The authors have clarified all my concerns.

I am grateful for the explanation of why the radial hairs look different in experimental images showing collagen density versus simulated images, which show area densification ratio. It is now clear to me that these two measurements, though related, are not the same and therefore the radial hairs appear different. The images the authors attached in the response were very helpful, particularly the ones showing the same simulation side-by-side once showing densification and once simulated collagen fiber density.

The authors should consider including this (or a similar) figure as a supplementary figure and make a note of this distinction when they present results of figure 3 (e.g. after the sentence ending in line 338), as other readers will likely have the same misunderstanding that I had.

Besides this point, I have only minor suggestions for improvement of the manuscript text.

% Minor comments

Abstract: The later is possible  the latter is possible

Line 156: a left parenthesis "(" is missing

In response to comments from another reviewer, the authors added lines 185-198. This section currently reads disjointed from the rest of the text. To a reader that has not read the reviews, it will be unclear why these studies are relevant to discuss.

I suggest to do a minor rewrite to make the text fit better with the preceding paragraph, explaining the motivation for bringing up these additional studies. Do they provide additional evidence? If yes, the authors could reformulate along the lines of "Additional evidence for the unstable buckling of collagen fibers comes from [51,52]... etc etc"

It is also not clear why reference 54 is discussed in this section. Is it an example for the analytical prediction? Mentioning this explicitly will help guide readers.

Line 190: incosistent  inconsistent

%

Overall, it is a very nice paper!

Reviewer #3: The authors addressed my comments well.

**Have the authors made all data and (if applicable) computational code underlying the findings in their manuscript fully available?**

Reviewer #1: Yes

Reviewer #2: Yes

Reviewer #3: Yes

PLOS authors have the option to publish the peer review history of their article (what does this mean?). If published, this will include your full peer review and any attached files.

Reviewer #1: No

Reviewer #2: No

Reviewer #3: No

---

## [Editor Report · Acceptance letter]

25 Jun 2024

PCOMPBIOL-D-23-01183R2 

Compressive Instabilities Enable Cell-Induced Extreme Densification Patterns in the Fibrous Extracellular Matrix: Discrete Model Predictions

Dear Dr Kalaitzidou,

I am pleased to inform you that your manuscript has been formally accepted for publication in PLOS Computational Biology. Your manuscript is now with our production department and you will be notified of the publication date in due course.

With kind regards,

Livia Horvath
